# Chromosome-scale assembly and analysis of biomass crop *Miscanthus lutarioriparius* genome

Jiashun Miao [1,2,6], Qi Feng [1,6], Yan Li[1], Qiang Zhao[1], Congcong Zhou[1], Hengyun Lu[1], Danlin Fan[1], Juan Yan[3], Yiqi Lu[1], Qilin Tian[1], Wenjun Li [1], Qijun Weng[1], Lei Zhang[1], Yan Zhao[1], Tao Huang [1], Laigeng Li [1], Xuehui Huang [4], Tao Sang[5✉] & Bin Han [1✉]

*Miscanthus*, a rhizomatous perennial plant, has great potential for bioenergy production for its high biomass and stress tolerance. We report a chromosome-scale assembly of *Miscanthus lutarioriparius* genome by combining Oxford Nanopore sequencing and Hi-C technologies. The 2.07-Gb assembly covers 96.64% of the genome, with contig N50 of 1.71 Mb. The centromere and telomere sequences are assembled for all 19 chromosomes and chromosome 10, respectively. Allotetraploid origin of the *M. lutarioriparius* is confirmed using centromeric satellite repeats. The tetraploid genome structure and several chromosomal rearrangements relative to sorghum are clearly demonstrated. Tandem duplicate genes of *M. lutarioriparius* are functional enriched not only in terms related to stress response, but cell wall biosynthesis. Gene families related to disease resistance, cell wall biosynthesis and metal ion transport are greatly expanded and evolved. The expansion of these families may be an important genomic basis for the enhancement of remarkable traits of *M. lutarioriparius*.

[1] National Center for Gene Research, State Key Laboratory of Plant Molecular Genetics, CAS Center of Excellence in Molecular Plant Sciences, Institute of Plant Physiology and Ecology, Chinese Academy of Sciences, Shanghai 200233, China. [2] University of Chinese Academy of Sciences, Beijing 100049, China. [3] CAS Key Laboratory of Plant Germplasm Enhancement and Specialty Agriculture, Wuhan Botanical Garden, Chinese Academy of Sciences, Wuhan, Hubei 430074, China. [4] College of Life Sciences, Shanghai Normal University, Shanghai 200234, China. [5] Key Laboratory of Plant Resources and Beijing Botanical Garden, Institute of Botany, Chinese Academy of Sciences, Beijing 100093, China. [6]These authors contributed equally: Jiashun Miao, Qi Feng. ✉email: sang@ibcas.ac.cn; bhan@ncgr.ac.cn

The genus *Miscanthus* contains approximately 20 species, among which 7 species are widely distributed in China[1,2]. It is a rhizomatous perennial plant that has great potential for bioenergy production for its high biomass yield and strong stress tolerance[1]. It is one of a few $C_4$ photosynthetic plants that adapt to the cold conditions, which makes *Miscanthus* a valuable genetic resource for sugarcane cold tolerance breeding[1,3]. *Miscanthus* has been shown to be tolerant to heavy metals, which makes it an excellent material for phytoremediation of heavy metal contaminated soil[4].

*Miscanthus lutarioriparius*, an endemic *Miscanthus* species in middle and low ranges of the Yangzi River of China, can grow up to 7 m tall and has the highest biomass production among four major *Miscanthus* species that widely distributed in China[2,5]. It is found to have good paper-making properties, which is one way to change the shortage of raw materials for paper-making industry in China. The species is considered to be a promising second-generation energy crop because of its high biomass production and strong adaptability through high photosynthetic rates, water use efficiency and strong tolerance to drought and salt when growing on the marginal land[6,7].

*Miscanthus* is self-incompatible (SI), which may contribute to the plenty of genetic diversity and extensive environmental adaptation of the genus[1]. However, the heterozygosity in *Miscanthus* genome hindered the advances of genome sequencing and assembly in the past. The lack of genomic resources limited our understanding of the genomic basis of distinctive traits, genome evolution and the process of molecular breeding in *Miscanthus*. The ploidy and genome size of species in genus *Miscanthus* are quite different[8,9]. For instance, *Miscanthus sacchariflorus* could be with diploid or tetraploid[9,10]. Here, *M. lutarioriparius* ($2n = 2x = 38$) possesses a relatively large genome size with abundant repetitive sequences, which made it difficult to achieve a high-quality genome assembly using short-read sequencing technologies. Recently, the single-molecular, real-time sequencing technologies and high-throughput chromosome conformation capture (Hi-C) technology have been used in combination to make significant advances in the assembly of large and complicated plant genomes[11].

Here, we generate a chromosome-level reference genome of *M. lutarioriparius* by combining Oxford Nanopore sequencing and Hi-C technologies. This chromosome-level genome assembly of *M. lutarioriparius* together with the assembly of *Miscanthus sinensis*[12] will facilitate the better utilization of *Miscanthus* genetic resources in multiple aspects.

## Results

**High-quality genome assembly of *Miscanthus lutarioriparius*.** Before the genome de novo assembly, genome survey based on k-mer frequency revealed a high level of repeat content (~67.30%) in *M. lutarioriparius* (Supplementary Table 1 and Supplementary Fig. 1). The genome size estimated by k-mer statistics is about 2.19 Gb, close to 2.15 Gb determined using flow cytometry[13].

The genome of *M. lutarioriparius* was de novo assembled using the long reads generated by PromethION platform of Oxford Nanopore Technologies and incorporated with Hi-C technology for scaffolding. We generated 307.71 Gb Nanopore long reads with a read N50 length of 32.21 kb (Supplementary Tables 2 and 3, and Supplementary Fig. 2). For draft assembly improvement, three Illumina paired-end libraries with different insert-size were constructed, and a total of 205.74 Gb data were generated using Illumina HiSeq platform (Supplementary Table 4). The total length of contigs assembled by SMARTdenovo software is about 2250.39 Mb with contig N50 size of 1.71 Mb, which is a little larger than the estimated genome size. Sequence comparison of the assembled initial contigs to three Bacterial Artificial Chromosome (BAC) sequences determined by Sanger sequencing showed good agreement with high sequence identity, ranging from 99.52% to 99.68% (Supplementary Fig. 3). Redundant sequences resulting from genome heterozygosity were collapsed during the following Hi-C anchoring process, where the total length of assembly reduced from 2250.39 Mb to 2074.63 Mb (without gaps and Ns). The final genome assembly consists of 919 scaffolds, which spans 2074.80 Mb in total, with the scaffold N50 size of 113.46 Mb (Table 1). The final assembly covers 96.64% of 2147 Mb genome size that was estimated by flow cytometry[13]. About 94.30% of total sequences were anchored into 19 pseudochromosomes with sizes ranging from 61.78 Mb to 150.81 Mb (Fig. 1a and Supplementary Table 5). Syntenic blocks account for 87.36% of total coverage across the whole-genome assembly (Fig. 1b and Supplementary Table 6).

To evaluate the reliability of the Hi-C scaffolding, we employed two Hi-C scaffolding programs LACHESIS[14] and 3D de novo assembly (3d-dna) pipeline[15] to anchor the contigs, respectively. The scaffolds constructed by two methods were massively consistent in contig order and orientation. LACHESIS and 3d-dna pipeline anchored and oriented 1956.46 Mb (94.30%) and 1806.79 Mb (93.93%) contigs into 19 pseudochromosomes, respectively (Fig. 1a and Supplementary Fig. 4). Except that Chr15 and Chr19 have relatively lower mean sequence identity (~85%), rest of the scaffolds show very high sequence identity between two methods, providing confidence in the scaffolding results. Several small inversions were observed at the end of chromosomes Chr09, Chr13, and Chr15 when comparing two results (Supplementary Fig. 5). The LACHSIS assembly was selected as the final genome assembly, since it is better than 3d-dna assembly by further quality evaluation (Supplementary Table 7).

We further verified the quality of the Hi-C scaffolding by analyzing the genomic synteny between *M. lutarioriparius* and

**Table 1 Statistics of genome assembly.**

|  | Hi-C scaffolds |
|---|---|
| **Assembly features** | |
| Number of scaffolds | 919 |
| Total size of scaffolds | 2074.80 Mb |
| Longest scaffold | 150.81 Mb |
| Shortest scaffold | 16.94 kb |
| Mean scaffold size | 2.26 Mb |
| Median scaffold size | 130.09 kb |
| N50 scaffold length | 113.46 Mb |
| L50 scaffold count | 8 |
| Scaffold GC content | 45.46% |
| Scaffold N content | 0.01% |
| Percentage of assembly in scaffolded contigs | 94.30% |
| Average number of contigs per scaffold | 2.9 |
| BUSCO (complete) | 97.40% |
| LTR Assembly Index (LAI) | 12.11 |
| **Gene models** | |
| Number of gene models | 68,328 |
| Mean coding sequence length | 1215 bp |
| Mean number of exons per gene | 4.77 |
| Mean exon length | 255 bp |
| Mean intron length | 727 bp |
| **Non-protein-coding genes** | |
| Number of miRNA gene | 521 |
| Number of tRNA gene | 1164 |
| Number of rRNA gene | 257 |
| Number of snoRNA gene | 970 |
| Number of snRNA gene | 97 |

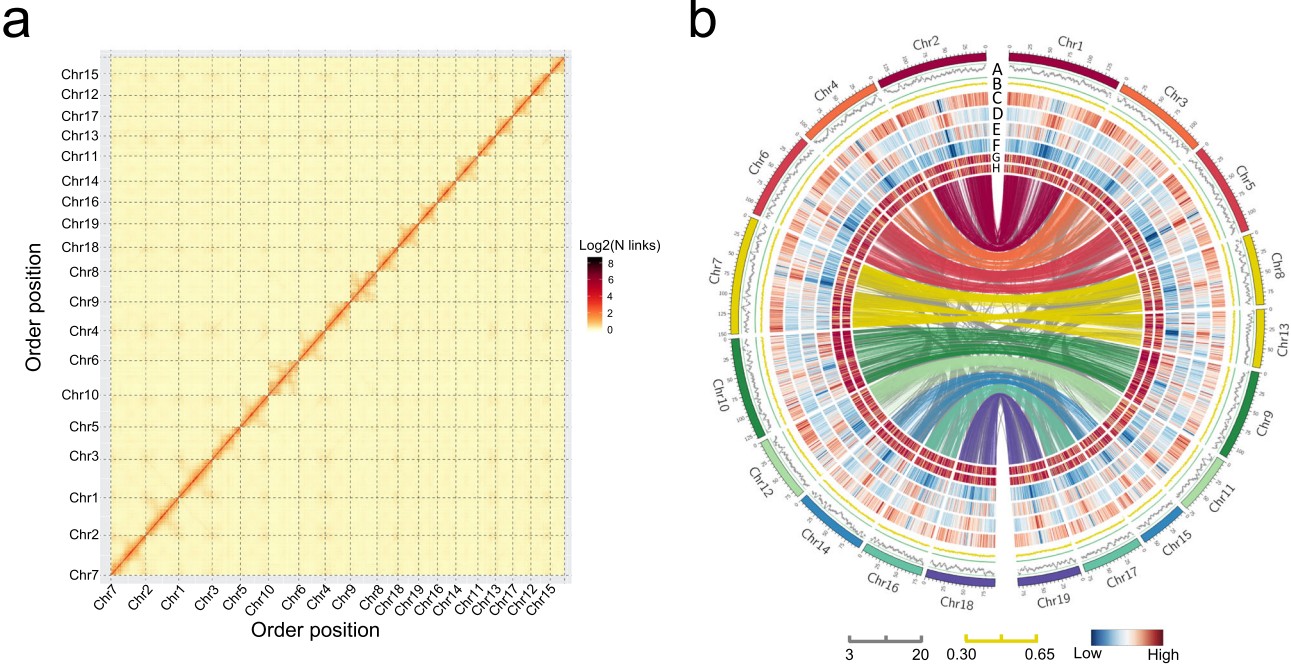

**Fig. 1 Overview of the *M. lutarioriparius* genome assembly. a** Genome-wide Hi-C map of *M. lutarioriparius*. Post-clustering heatmap shows density of Hi-C interactions between contigs from LACHESIS pipeline. **b** Circos plot of *M. lutarioriparius* genome assembly. The outermost layer of colored blocks is a circular representation of the 19 pseudochromosomes, with scale mark labeling each 5 Mb. The LTR Assembly Index (LAI) score is shown in track (A). The density of GC content (B), gene (C), *Gypsy* (D), *Copia* (E), DNA TE (F), transcriptome expression level of rhizome (G), and lateral bud (H) were calculated using 100 kb non-overlap window. The innermost layer shows inter-chromosomal synteny, with colored upper-layer links representing syntenic blocks generated from the recent *M. lutarioriparius* WGD, and the gray low-layer links representing homologs generated from older duplications.

sorghum genome. The results showed a conserved synteny in both coding sequences (CDSs) and genomic scale (Fig. 2a and Supplementary Fig. 6). The LAI score of assembly constructed by Hi-C scaffolding was 12.11 (Supplementary Fig. 7), reaching to the criterion of reference quality[16]. A total of 1339 (97.4 %) complete gene models among 1375 conserved genes in Benchmarking Universal Single-Copy Orthologs (BUSCO) were recalled (Table 1). Three Illumina DNA libraries were mapped back to the assembly, and overall alignment rates were 99.82%, 99.79%, and 99.78%, respectively. RNA sequencing data generated for genome annotation was also used to assess the quality of assembly by mapping back to the assembly, and overall alignment rates ranged from 82.55% to 95.37% in nine libraries (Supplementary Table 8). Therefore, the genome assembly of *M. lutarioriparius* presents high contiguity and sequence quality.

The average GC content of *M. lutarioriparius* genome is about 45.46% (Supplementary Table 9). Relatively lower GC content was observed for chromosome 9 and 10 (Supplementary Fig. 8). The potential centromeric sequences of *M. lutarioriparius* were assembled for all 19 chromosomes (Supplementary Fig. 9 and Supplementary Table 10). The centromeric satellite sequences of *M. lutarioriparius* consist of 137 bp highly repetitive units and the sizes vary from 106.8 kb (chromosome 19) to 7.7 Mb (chromosome 10). Telomere sequence of chromosome 10 was assembled in our assembly, and 1917 *Arabidopsis*-type 7-bases TTTAGGG[17] repeats were detected (Supplementary Fig. 9).

**Genome annotation and repeat elements in *Miscanthus lutarioriparius*.** A total of 68,328 gene models were predicted, with an average CDS length of 1215 bp, of which about 94.75% (64,742) were assigned to 19 chromosomes (Supplementary Table 11). The length distribution of CDSs, exons, and introns of *M. lutarioriparius* are similar to that of *Saccharum spontaneum* and *Sorghum bicolor* (Supplementary Fig. 10), providing confidence in

the gene prediction results. A total of 4031 transcription factors were annotated. In addition, 1164 tRNA genes, 257 rRNA genes, 521 microRNA genes (miRNAs), 98 small nuclear RNA genes (snRNAs), and 970 small nucleolar RNA genes (snoRNAs) were predicted by integrating different methods (Table 1).

We evaluated the completeness of gene prediction with 1375 BUSCO genes from embryophyta_odb10, of which 1350 BUSCOs (98.2%) were complete (Supplementary Table 12). Among 68,328 predicted proteins, 63,076 (92.31%) were classified by InterProScan[18], 63,590 (93.07%) were classified by eggNOG-mapper[19], 22,902 (33.52%) were functional annotated in KEGG database, and 39,173 (57.33%) were assigned with Gene Ontology (GO) annotation.

The average GC content of the CDSs of *M. lutarioriparius* (56.40%) is similar to that of *S. spontaneum* (56.40%), *S. bicolor* (56.74%), and *Setaria italica* (56.50%), but higher than that of *Arabidopsis thaliana* (44.45%) (Supplementary Table 13). The GC content and GC3s (GC of silent 3rd codon position) of CDSs showed a bimodal distribution in *M. lutarioriparius* (Supplementary Figs. 11 and 12). Previous studies demonstrated that the various GC content of genome region may have different biological properties, such as gene density, composition of repeat sequences, and recombination[20,21]. Strong positive linear correlation (Pearson correlation coefficient = 0.95, FDR = 8.25e-10) was detected between GC content and gene density in *M. lutarioriparius* (Supplementary Fig. 13). Notably, gene density of chromosome 9 (25.19 genes/Mb) and 10 (26.27 genes/Mb) are significantly lower than the average gene density of whole genomes (32.93 genes/Mb).

A total of 1.34 Gb (64.39% of nucleus genome) interspersed repeats were identified in *M. lutarioriparius* genome (Supplementary Table 14). The long-terminal repeat retrotransposon (LTR-RT) is the most abundant type of repetitive sequences in *M. lutarioriparius*, spanning 46.78% of the nucleus genome. A total

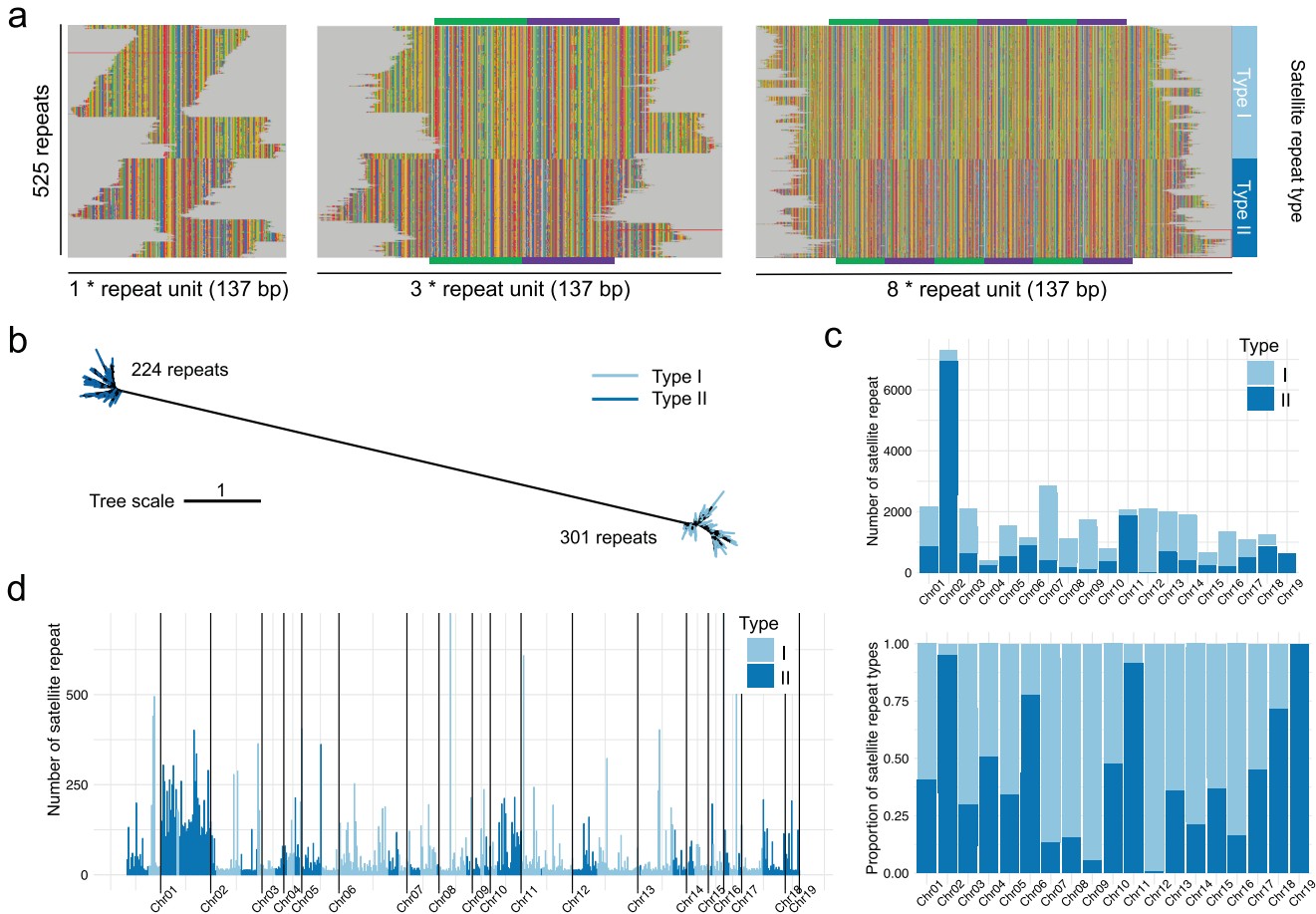

**Fig. 2 Evolution of satellite repeats of *M. lutarioriparius*. a** Multiple sequence alignment of 525 satellite repeats derived from screening the *M. lutarioriparius* genome sequence. One, three, and eight tandem satellite repeats were used to perform alignment, respectively. Here, 301 sequences were clustered into Type I satellite repeats and 224 into Type II satellite repeats. **b** Maximum likelihood tree of 525 satellite repeats. **c** Absolute and relative frequency distributions of Type II satellite repeats of each chromosome. **d** Copy number distribution of 525 satellite repeats, which is arranged according to the order on chromosomes. Source Data are provided as a Source Data file.

of 8848 intact LTR-RTs were classified, and the frequency distribution of insertion time showed a burst of LTR-RTs 1–2 Ma (million years ago) (Supplementary Fig. 14). The largest LTR-RTs superfamily *Gypsy*, comprising ~35.2% of the genome, is concentrated near the putative centromeres (Supplementary Fig. 15d). The second largest superfamily of LTR-RTs *Copia* comprises ~11.6% of the genome (Supplementary Fig. 15c). Other interspersed repeats, LINEs (Long Interspersed Nuclear Elements) and SINEs (Short Interspersed Nuclear Elements) occupy 1.21% and 0.16% of the genome, respectively (Supplementary Fig. 15a, b). The sequence divergence rate of LINEs is relatively higher (median = 20.6%) compared to other types of transposable elements (Supplementary Fig. 16). DNA transposons comprises ~9.64% of the *M. lutarioriparius* genome. Strikingly, DNA transposons shows obvious enrichment on the arms of the GC-poor chromosome 9 and 10 (Fig. 1b and Supplementary Fig. 15e, f). Whereas, there is no significant correlation between the DNA transposons content and GC content for both chromosome 9 and 10. A total of 19,062 miniature inverted-repeat transposable elements (MITEs) were identified, spanning 0.23% of the genome (Supplementary Table 15 and Supplementary Fig. 17). In total, 517,973 tandem repeats were identified, spanning 3.74% of the genome (Supplementary Table 16 and Supplementary Fig. 9).

**Centromeric evolution in *Miscanthus lutarioriparius*.** Two types of centromeric satellite repeats were identified in

*M. lutarioriparius*, supporting the allotetraploid origin of *M. lutarioriparius* (Fig. 2). The satellite monomers of each type are arranged in large tandem arrays (Fig. 2d). Except Chr19, two types of satellite repeats were observed in the same centromere region, indicating that there has been recombination between these two types of centromeres (Fig. 2c, d). Recently, a research in maize showed the birth and death of centromere after chromosomal rearrangement occurred in a very short timeframe, even a few cell cycles[22]. Only one centromere region was identified in *M. lutarioriparius* Chr07, implying the other one centromere probably inactivated after the chromosomal rearrangement event[23] (Fig. 2a and Supplementary Fig. 9).

**Genomic structure variation in *Miscanthus lutarioriparius*.** Investigation of collinear orthologs between *M. lutarioriparius* and sorghum confirmed the occurrence of the recent whole-genome duplication (WGD) in *M. lutarioriparius* (Fig. 3a). Except that chromosome 7 of *M. lutarioriparius* has syntenic relationship with two sorghum chromosomes (SbChr04 and SbChr07), rest of the chromosomes are aligned to one sorghum chromosome (Fig. 3a). There are 65,044 (95.19%) *M. lutarioriparius* genes having at least one syntenic gene in *S. biocolor*. About 87.05% (29,710) of sorghum genes have two syntenic genes in *M. lutarioriparius*, showing a predominant 2-to-1 pattern of syntenic depth between *M. lutarioriparius* and sorghum, and revealing the tetraploid nature of the diploid *M. lutarioriparius*

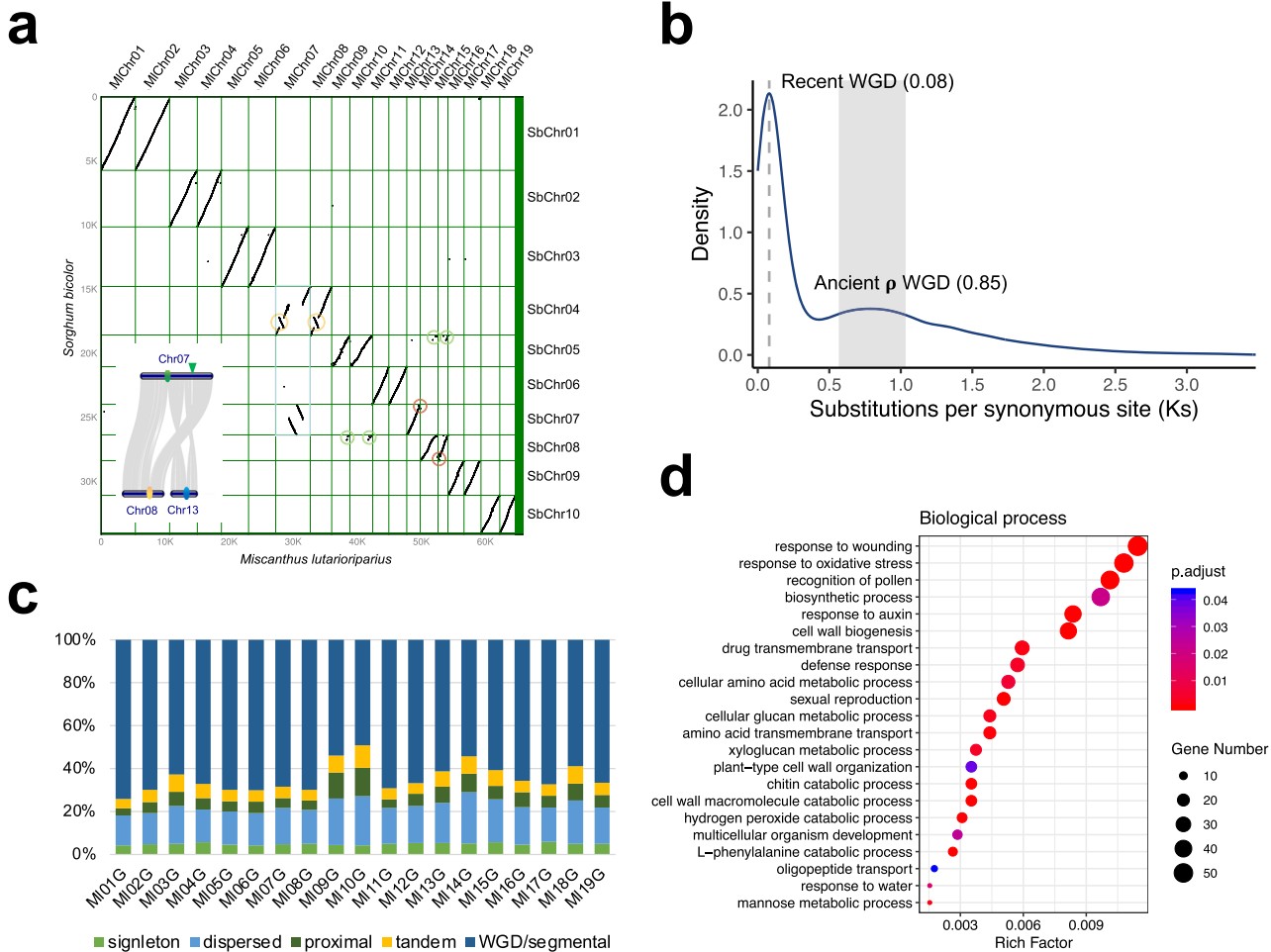

**Fig. 3 Whole-genome duplication events within *M. lutarioriparius* genome. a** Macrosynteny dotplot of *M. lutarioriparius* and *S. bicolor* chromosomes. The blue rectangle indicates the chromosome fusion that presumably occurred after the *M. lutarioriparius*-specific WGD. The green circles indicate that the chromosome ends of MlChr09/10 and MlChr14/15 are highly collinear. The orange circles indicate the intra-chromosomal inversions that probably happened before the *M. lutarioriparius*-specific WGD. The red circles indicate the intra-chromosomal inversions that may happened after the *M. lutarioriparius*-specific WGD. The ellipses in the small plot at the lower left represent the centromeric regions, and the green arrow indicates the disappeared centromere. **b** The frequency distributions of synonymous substitution rates (*Ks*) of homologous gene pairs that located in the collinearity blocks of *M. lutarioriparius* versus *M. lutarioriparius*. The numbers in parentheses indicate peak values of recent *M. lutarioriparius* WGD and the grass lineage shared ρ WGD. **c** Classification of gene duplicates origin in *M. lutarioriparius* genome. The origins of gene duplicates were classified into five types: whole-genome/segmental duplication (collinear genes in collinear blocks), tandem duplication (consecutive repeat), proximal duplication (two duplicated genes are distributed adjacent to each other on chromosomes, with no more than 10 genes spaced but not adjacent), dispersed duplication (duplication type other than WGD/segmental, tandem and proximal), and singleton (no duplication). **d** Gene Ontology (GO) enrichment analysis of tandem duplicated genes in *M. lutarioriparius* genome (biological process category). The color of circle represents the FDR (false discovery rate) in the hypergeometric test corrected using BH (Benjamini and Hochberg) method. The size of circle represents the gene count of the GO terms. Source Data are provided as a Source Data file.

(Fig. 3a). In all, 2802 (8.21%) sorghum genes have more than two syntenic genes in *M. lutarioriparius* genome, which presumably resulted from the segmental, tandem or single-gene duplication that occurred in *M. lutarioriparius* after it split with sorghum. Only about 0.68% (233) of genes in sorghum have no syntenic gene in *M. lutarioriparius*, indicating that few genes were lost in *M. lutarioriparius* after it split with sorghum.

By comparing genomes of *M. lutarioriparius* and *S. bicolor*, several chromosome rearrangements in *M. lutarioriparius* were identified (Fig. 3a and Supplementary Fig. 6). Two large inversions were observed on the two homologous chromosomes of *M. lutarioriparius* (MlChr07 and MlChr08), which is consistent with the research carried out in *M. sinensis* based on the RNA-Seq data[23] (Fig. 3a). The common ancestor's chromosomes of SbChr04 and SbChr07 were fused into one chromosome

MlChr07 in *M. lutarioriparius*, which probably occurred after the *M. lutarioriparius*' recent WGD, resulting in MlChr07 aligning to two sorghum chromosomes (Fig. 3a). The ends of MlChr09/10 and MlChr14/15 are highly collinear, which is common to major Poaceae lineages[24,25] (Fig. 3a).

For dating the recent WGD in *M. lutarioriparius*, genomic synteny analysis based on the self-comparison of *M. lutarioriparius* protein-coding genes was carried out. The two peaks of the synonymous substitution values' (*Ks*) frequency distribution indicate that an ancient burst gene duplication occurred prior to the recent WGD (Fig. 3b). The calculation of *Ks* for *M. lutarioriparius* versus *M. lutarioriparius* collinear gene pairs suggested that the recent WGD event may date back to ~6.15 Ma (Fig. 3b).

We analyzed the duplicate gene origins in *M. lutarioriparius*. The results suggest that the WGD/segmental duplication is the

predominant type of gene duplication (63.96%, 43,704) compared to the other three types: dispersed duplication (18.59%, 12,700), tandem duplication (6.39%, 4,365), and proximal duplication (6.24%, 4267) (Supplementary Fig. 18a). *Miscanthus* has the highest proportion of tandem duplication origin compared to the other eight investigated taxa (Supplementary Fig. 18b). The chromosome 9 and 10 have the highest proportion of proximal and tandem duplication compared to that of other chromosomes (Fig. 3c). The analysis of GC3s and GC content of genes of different duplication origins indicates that the tandem duplicates have the highest GC3s value and lowest effective number of codons (ENC/Nc) (Supplementary Fig. 19).

To find out if genes of different duplication origins have biological function preference, functional enrichment analysis was carried out. Genes created through WGD/segmental duplication were enriched with the GO terms like 'cell redox homeostasis', 'intracellular protein transport', 'signal transduction', 'metal ion transport', and 'photosynthesis' (Supplementary Fig. 20). Previous study has demonstrated that the tandem arrayed genes in rice and *Arabidopsis* genomes were enriched in the function of 'biotic and abiotic stress'[26]. While gene duplicates originated from tandem duplication in *M. lutarioriparius* were enriched in the GO terms related not only to biotic and abiotic stress response, but also to cell wall biosynthesis (Fig. 3d and Supplementary Figs. 21 and 22). Tandemly arrayed genes are thought to be volatile after gene duplication, so the retained tandem genes may indicate functional importance[27]. These significantly enriched GO terms of tandem gene duplicates were closely associated with the distinctive features of *M. lutarioriparius*: higher environmental adaptability, disease resistance, and lignocellulosic biomass. It is noteworthy that a GO term named 'recognition of pollen', also known as 'self-incompatibility', was significantly enriched in tandem duplicates, indicating the importance of self-incompatibility in *M. lutarioriparius* (Fig. 3d). The duplicate genes that resulted from proximal duplication were mainly enriched in the function of cell wall biosynthesis, such as 'transferring hexosyl groups' and 'polysaccharide binding' (Supplementary Fig. 23). These gene duplicates classified as dispersed were enriched in the functions like 'DNA repair', 'telomere maintenance', 'nucleosome assembly', 'protein heterodimerization activity', and 'DNA helicase' (Supplementary Fig. 24).

**Genome evolutionary history and genetic diversity analysis of *Miscanthus lutarioriparius*.** To investigate the genome evolutionary history of *M. lutarioriparius*, gene family clustering was carried out using *M. lutarioriparius* and seven other angiosperm species. From the gene family clustering, 122 single-copy orthologs shared by *M. lutarioriparius* and seven other plants were used for phylogenetic reconstruction and species divergence time estimation, which showed *M. lutarioriparius* was closest relative to *S. spontaneum*[28]. We estimated that *M. lutarioriparius* and *S. spontaneum* shared a common ancestor ~7.97 Ma. The divergence of *M. lutarioriparius* and *S. bicolor* was estimated to occur ~9.59 Ma (Fig. 4b).

Transcriptome data of 79 individuals from 10 populations of *M. lutarioriparius* were used to investigate the genetic diversity of *M. lutarioriparius*[29]. A total of 3,209,041 SNPs and 279,810 InDels were recalled in these transcriptomes. Phylogenetic analysis based on SNPs clustered all individuals into two groups, and the genetic variation within some populations was greater than that between populations (Supplementary Fig. 25a). Population structure analysis and principal component analysis further confirmed *M. lutarioriparius* in China could be classified into two groups (Supplementary Fig. 25b, c).

**Comparative genomics of gene family in *Miscanthus lutarioriparius*.** To reveal the genomic basis of *M. lutarioriparius*' distinctive phenotypes, we investigated the evolution of gene families by characterizing unique and shared genes' families among different plants. Through comparing gene families of seven grass subfamilies, including Pooideae (*Brachypodium distachyon*), Ehrhartoideae (*Oryza sativa*), Panicoideae (*Zea mays*, *Sorghum bicolor*, *Setaria italica*, *Saccharum spontaneum*, and *Miscanthus lutarioriparius*), and one dicot plant (*Arabidopsis thaliana*), we characterized 57,710 genes of *M. lutarioriparius* into 21,515 gene families. Further comparison of the species in the subfamily Panicoideae, including *M. lutarioriparius*, *Z. mays*, *S. bicolor*, *S. italica*, and *S. spontaneum*, revealed 16,080 gene families distributed among all five genomes and 144 gene families that were unique to *M. lutarioriparius* (Fig. 4a). Ten of these 144 gene families contain NB-ARC domain, a signaling motif shared by most plant resistance (*R*) gene productions[30], implying that stronger disease resistance might have evolved in *M. lutarioriparius*. And three of these 144 gene families were annotated to be cytochrome P450, a multifunctional oxidase involved in the biosynthesis of defensive compounds and hormones in plants[31]. GO enrichment analysis of these 144 gene families revealed totally six significant GO terms, including 'peroxidase activity', 'response to oxidative stress', 'ubiquitin-like modifier activating enzyme activity', 'cellular amino acid metabolic process', and 'DNA integration and carboxy-lyase activity' (Fig. 4c).

Gene family expansion and contraction analysis for *M. lutarioriparius* classified 9509 and 3228 gene families as expanded and contracted, respectively (Fig. 4b), among which, 219 were identified as rapidly evolving families (211 expanded and 8 contracted). These rapidly evolving gene families might give us some hints about adaptability of *M. lutarioriparius* to the harsh environment. Functional annotation of these 211 rapidly expanded gene families revealed that they were involved in a variety of plant biological processes, but most notably in dealing with diverse biotic and abiotic stresses, particularly disease resistance (Fig. 4d). Among these 211 rapidly expanded gene families, 21 gene families consisting of 334 genes contained the NB-ARC domain. Four rapidly expanded gene families consisting of 56 genes were annotated to be related to the xylanase inhibitor, which could be induced by biotic and abiotic stimulus, and involve in plant defense against fungal pathogens[32]. Other stress-resistance-related functional annotation for these rapidly evolving gene families included 'salt stress response/antifungal', 'wound induced protein', 'ubiquitin family', 'WRKY DNA-binding domain', 'Cytochrome P450', 'thaumatin family', and 'terpene synthase' (Fig. 4d).

GO enrichment analysis for the 9509 expanded gene families revealed that the significantly enriched GO term with the largest gene members (155) in biological process category was 'metal ion transport' (Fig. 4e). Notably, GO terms related to the cell wall biosynthesis were present in the enrichment results (Fig. 4e and Supplementary Fig. 26).

Most of the disease resistance genes cloned in plants are found to encode nucleotide-binding-site-leucine-rich-repeat (NBS-LRR) proteins[33]. In *M. lutarioriparius* genome, 547 NBS-LRR coding genes were identified, which is more than that of rice (~500)[34], sorghum (211–346)[24,35], maize (137)[35], and *S. spontaneum* (361)[28] (Supplementary Data 1). *M. lutarioriparius* NBS-LRR genes mostly encode the CC-NBS-LRR (262). A total of 234 (42.8%) NBS-LRR genes are enriched on chromosome 9 and 10, most of which fall into multi-gene clusters, supporting the conservation of *R* gene location[24] (Supplementary Fig. 27c). About 17.4% (95) NBS genes were classified into tandem duplicates, which was significantly higher than that at the genome-wide level (6.39%), indicating the important role of

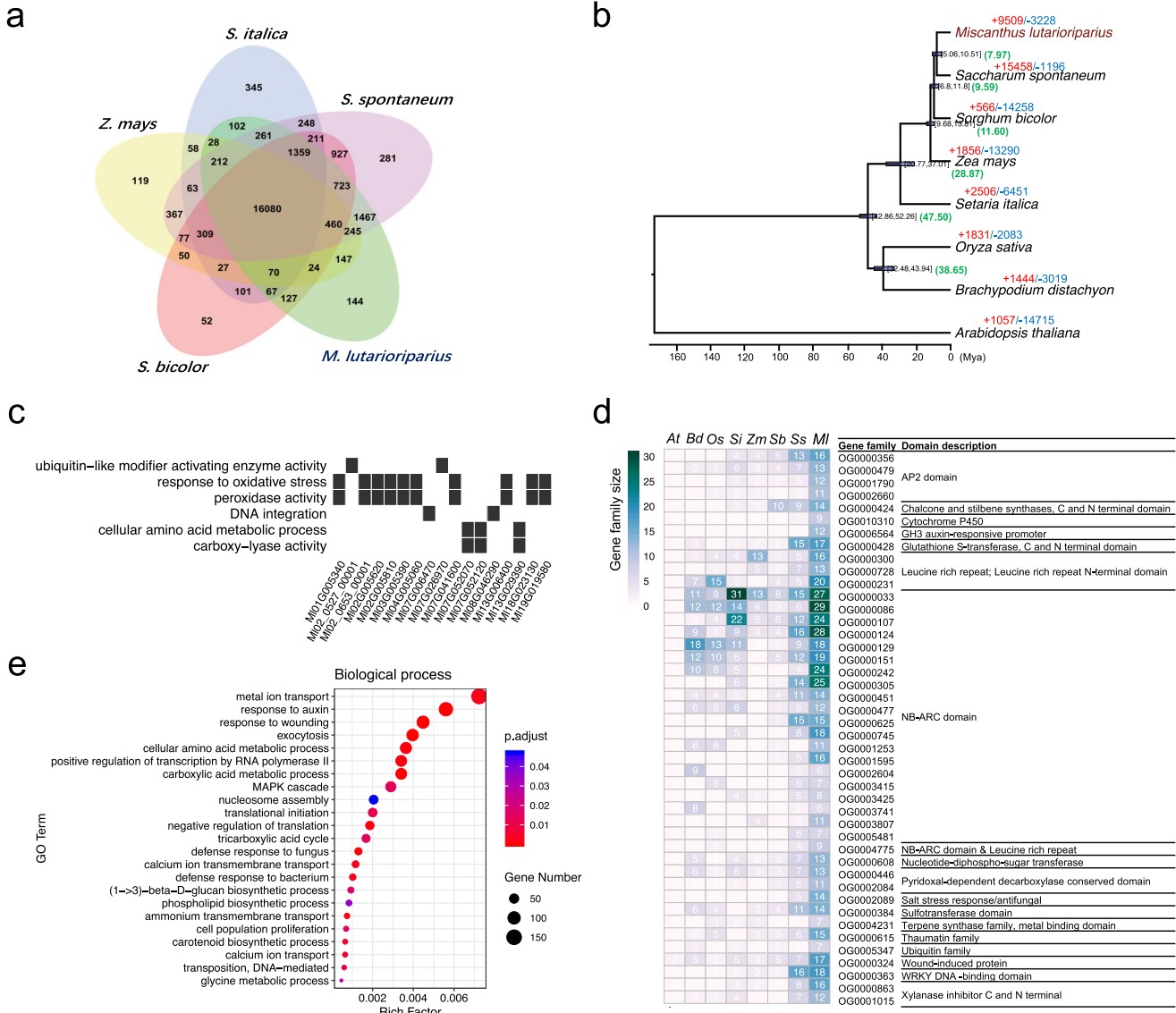

**Fig. 4 Comparative genomics of gene family of *M. lutarioriparius*. a** Venn diagram of shared orthologous genes' families among five Panicoideae species genomes. **b** Inferred phylogenetic tree reconstructed using 122 single-copy ortholog genes shared by 8 species identified using OrthoFinder. Divergence timings were indicated using transparent blue bars at the internodes with 95% highest posterior density (HPD). The number of expanded and contracted gene families was marked with plus and minus ahead the digitals, respectively. **c** GO enrichment results of 144 *M. lutarioriparius*-specific gene families. **d** Functional annotation for the rapidly evolving (expanded) gene families in *M. lutarioriparius*. Left panel shows the gene family size among 8 species and right panel shows the functional annotation for gene families. **e** GO enrichment results of 9509 expanded gene families of *M. lutarioriparius* (biological process category). Source Data are provided as a Source Data file.

tandem duplication in the expansion of NBS genes in *M. lutarioriparius*. Phylogenetic analysis based on NBS domain showed no distinct clades, and yielded a 'star-like' topology (Supplementary Fig. 27), which was similar to that of rice and sorghum[35].

Complex carbohydrates of plants are the important food sources and renewable biofuel feedstock. Carbohydrate-active enzymes (CAZymes) play critical role in complex carbohydrate metabolism[36]. A total of 2919 (4.3%) genes encoding CAZyme domains were identified in *M. lutarioriparius*, which was the most abundant among the 12 species investigated (Supplementary Fig 28). GTs comprise approximately 46% of the CAZyme domain content in *M. lutarioriparius*, with GHs having a relative frequency of 32%. CEs have a smaller proportion (4.6%) compared to other species. A total of 47 GT families are present in *M. lutarioriparius*, among which 12 GT families (GT1, GT2,

GT47, GT31, GT8, GT0, GT61, GT106, GT4, GT77, GT37, and GT34) account for more than 80% of all GT members. GT1 is the largest GT family with 396 (27.54%) members in *M. lutarioriparius*, and GT1 subfamily also constitute the majority of GT genes in *Arabidopsis* (>25%) and rice (>35%)[37] (Supplementary Fig. 29). In addition, 27 genes coding cellulose synthases (CesA) were identified in *M. lutarioriparius* (Supplementary Table 17). Transcriptome analysis showed almost half of *CesA* genes have specific high expression in the middle part of internode (Supplementary Fig. 30b), implying that the cellulose synthesis may be more active in the middle part of internode. A total of 90 genes encoding cellulose synthase-like (Csl) were identified in *M. lutarioriparius*, which was larger than that of maize (33), rice (35), and *Arabidopsis* (29) (Supplementary Data 2). Phylogenetic analysis separated all *M. lutarioriparius* Csl genes into six distinct groups and revealed that *CslD* group is most closely related to

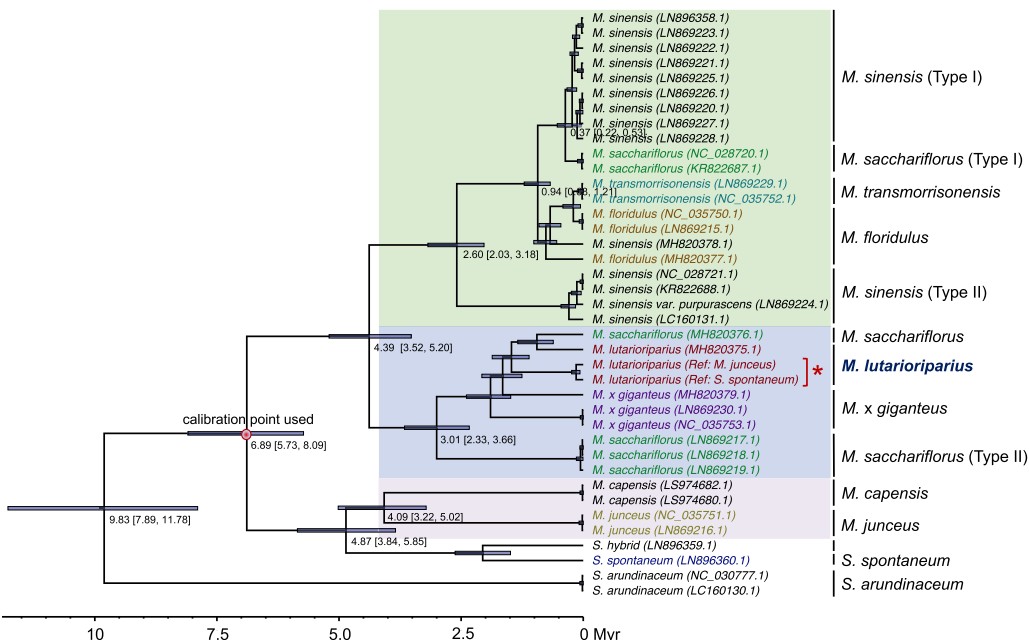

**Fig. 5 Time-calibrated phylogeny analysis based on the chloroplast genomes.** The calibration point was indicated using red point (data source: http://www.timetree.org/). The light blue bars at nodes correspond to the 95% highest posterior density (HPD) of the age of the nodes. The bottom axis is in millions of years (Myr). The NCBI Reference IDs were indicated in the parentheses. Three background colors were used to indicate distinct groups in genus *Miscanthus*. The red star was used to indicate the chloroplast genome assemblies of our sequencing accession, which were reconstructed directly from the Illumina paired-end reads derived from whole-genome sequencing using a baiting and iterative mapping approach, by which two references were used (*M. junceus* and *S. spontaneum*).

*CesA* family, but *CslD* and *CesA* differed greatly in gene structure (Supplementary Fig. 31). Tandem, WGD/segmental and proximal duplication greatly expanded the gene families of *Csl* and *CesA* in *M. lutarioriparius* (Supplementary Figs. 30–32). So, we speculated that the higher cellulose content of *M. lutarioriparius* may be contributed by the expansion of the cellulose-synthase-related gene families (Supplementary Table 18).

Lignin, a component of plant cell walls, is an amorphous polymer that is responsible for maintaining the rigidity and mechanical structure of cell walls, providing a barrier against pathogens. Lignin can directly reduce the entry of heavy metals into the root system, enhancing the tolerance of plants to heavy metals[38]. In *M. lutarioriparius*, 333 genes belonging to 10 gene families were inferred to involve in lignin biosynthesis, which is greater than that of sorghum (141) and rice (155)[39] (Supplementary Data 3). Various types of gene families are distributed closely on chromosomes, which may be beneficial for the improvement of the lignin biosynthesis efficiency (Supplementary Fig. 33a).

**C$_4$ pathway genes of *Miscanthus lutarioriparius*.** Genes involved in the C$_4$ photosynthesis pathway of *M. lutarioriparius* were identified on the basis of homology to C$_4$ pathway orthologs in sorghum. *M. lutarioriparius* contains 55 putative genes involved in nicotinamide adenine dinucleotide phosphate-malic enzyme (NADP-ME) C$_4$ pathway carbon fixation, including 10 carbonic anhydrase (CA) encoding genes, 12 phosphoenolpyruvate carboxylase (PEPC) encoding genes, six phosphoenolpyruvate carboxylase kinase (PPCK) encoding genes, three pyruvate orthophosphate dikinase (PPDK) encoding genes, six pyruvate phosphate dikinase regulatory protein (PPDK-RP) encoding genes, 13 NADP-linked malic enzymes (NADP-ME) encoding genes, three NADP-linked malate dehydrogenase (NADP-MDH), and two RbcS encoding genes (Supplementary Note 1, Supplementary Tables 19–26, and Supplementary Figs. 34–50).

Most C$_4$ photosynthesis genes in *M. lutarioriparius* were mainly expanded through the WGD, leading to the number of them almost twice that of sorghum. Proximal, tandem, and segmental duplication further expanded the C$_4$ photosynthesis gene families in *M. lutarioriparius* (Supplementary Note 1). Most non-C$_4$ isoforms showed very low expression level among all transcriptome samples compared to these C$_4$ isoform genes. The C$_4$ isoform gene duplicates created by recent WGD showed a similarly high expression in leaf sample, implying that both gene duplicates probably play a role in the C$_4$ photosynthesis and contribute to *Miscanthus* C$_4$ photosynthesis at low temperature. While some non-C$_4$ isoforms of PPCK family showed distinctive expression patterns (Supplementary Note 1), for instance, the non-C$_4$ isoform Ml12G018490 had the highest expression level in root sample. The variation of expression pattern may indicate the neofunctionalization followed the C$_4$ photosynthesis gene duplication.

**Phylogenetic analysis based on chloroplast genome.** A nearly complete chloroplast genome of *M. lutarioriparius* assembled using the Illumina short reads derived from whole-genome sequencing data. The assembled chloroplast genome is 142,989 bp in length, with 123 protein-coding, 71 tRNA, and eight rRNA genes annotated (Supplementary Fig. 51). Phylogenetic analysis by integrating other released chloroplast genomes in *Miscanthus* genus confirmed our sequencing sample belonging to *M. lutarioriparius* (Fig. 5, and Supplementary Figs. 52 and 53). Phylogenetic analysis based on chloroplast genome is of great value in providing support to evolutionary relationships for species in *Miscanthus* genus (Supplementary Data 4). The accessions we investigated were divided into three distinct groups, one group containing the accessions from *M. sinensis* (Type I and II), *M. sacchariflorus* (Type I), *M. transmorrisonensis*, and *M. floridulus*, one group containing the accessions from *M. lutarioripairus*, *M. sacchariflorus* (Type II), and *M. × giganteus* and

another group containing the accessions from *M. capensis*, *M. junceus*, and *S. spontaneum* (Fig. 5). The *M. sinensis* accessions were separated into two subgroups (Type I and II) distributed in two clades, suggesting higher genetic diversity in *M. sinensis*. All but one accessions of *M. sacchariflorus* were divided into two distant groups, with one closely related to the Type I *M. sinensis* accessions and the other one closely related to the accessions from *M.* × *giganteus* and *M. lutarioriparius*. *M.* × *giganteus* was thought to be the interspecific hybrid of diploid *M. sinensis* and tetraploid *M. sacchariflorus*, and the maternal lineage was inferred to be the tetraploid *M. sacchariflorus*[8]. So, we inferred these *M. sacchariflorus* accessions that closely related to *M.* × *giganteus* accessions were tetraploid. Our phylogeny analysis based on chloroplast genome sequences indirectly supports the view of point that tetraploid *M. sacchariflorus* and diploid *M. sacchariflorus* are taxonomically different[9]. In our phylogenetic analysis, *M. lutarioriparius* and one accession from *M. sacchariflorus* (MH820376.1) are the closest relatives. Strikingly, *M.* × *giganteus* accessions are most closely related to *M. lutarioriparius* and one accession of *M. sacchariflorus* (MH820376.1). To better uncover the complicated relationship in genus *Miscanthus*, more accessions, geographical origin, and karyotype need to be included in the phylogenetic analysis.

## Discussion

Here, we generated a qualified chromosome-scale assembly of the *M. lutarioriparius* genome using the long reads generated by Oxford Nanopore Technologies and Hi-C technology. The advance in genomic sequencing and assembly technologies makes it possible to produce chromosome-scale scaffolds in an accurate and cost-effective way. The chromosome-scale genome assembly of *M. lutarioriparius* provides crucial information for comparative genome studies for genus *Miscanthus*. We believe this study will contribute to the parsing of the genomic basis of distinctive traits displayed by *M. lutarioriparius* and the development of genome-assisted molecular breeding process.

The recent WGD and tetraploid genetic structure of diploid *M. sinensis* have been clearly demonstrated, and the chromosome 8 of *M. sinensis* was revealed to align to two sorghum chromosomes (7 and 8)[12,23,40]. Here, we found that *M. lutarioriparius* had a very similar genome structure to that of *M. sinensis* at the chromosome level, indicating that the two species diverged very recently (Supplementary Fig. 54a). The sequence similarity of CDSs between *M. lutarioriparius* and *M. sinensis* is about 98.59%.

We compared the sequence similarity between the two homoeologous chromosomes in *M. lutarioriparius*, relatively low values (31.66% to 33.47%) were present for all pairs (Supplementary Table 6). Swaminathan et al. suggested that polyploidizations in the lineages of multiple *Miscanthus* species were generated by the hybridization of closely related species in *Miscanthus* in allopolyploid fashion, by which their original chromosomal pairing patterns in the larger genome can be retained[23]. Here with the centromere sequences assembled in this study, we confirmed the allopolyploid origin of *M. lutarioriparius* at the molecular and chromosomal levels. By comparing *M. lutarioriparius* to *M. sinensis*, we assigned each chromosome in bulk to the A and B subgenomes (Supplementary Fig. 54b).

In the case of *M. lutarioriparius*, it is morphologically distinct from *M. sacchariflorus* and stands out as a taxon with the largest biomass production among *Miscanthus* species[2,5,9]. It also occupied a unique ecological niche, i.e., seasonally flooded riverbanks where other plant can hardly adapt[2]. It has also been shown as a valuable genetic resource for energy crop development based on its prominent performance of photosynthetic rates and water use efficiency when planted on the semiarid marginal land[6,7].

Phylogeny based on chloroplast genome sequence provided a perspective for the interspecific relationship of genus *Miscanthus*. Therefore, despite the existing controversies, the recognition of its species status would be valuable for its potential utilization as an important unit of genetic resource for bioenergy development.

We believe that our genome data will be helpful to functional genomic study and genetic improvement for *M. lutarioriparius*.

## Methods

**Plant material.** The *M. lutarioriparius* plant used in this study was sampled from Honghu Lake, Hubei Province, China, and transplanted in Wuhan Botanical Garden. The root tip was taken for karyotype and flow cytometry analysis to determine the genome size and diploidy of the plant. After that, the individual was propagated by asexual reproduction and brought to Shanghai, and transplanted in National Center for Gene Research, CAS.

**Oxford Nanopore sequencing library construction and sequencing.** High molecular weight (HMW) genomic DNA (gDNA) was extracted from young leaves with CTAB extraction method[41]. The gDNA was then purified with Qiagen Genomic-tip100/G (Cat ID: 10243, Qiagen, Germany) in accordance with the manufacturer's instructions. The DNA fragments with length ranging from 20 to 50 kb were selected using BluePippin Size-Selection System (Sage Science Inc., Beverly, MA, USA). The Nanopore libraries were then constructed using Oxford Nanopore LSK-109 kit following the manufacturer's instructions. Nanopore libraries were sequenced on PromethION platform with Flow cell R9.4.1 chemistry and guppy software; 307.71 Gb raw Nanopore data were generated. After adapter trimming and quality filtering, a total of 280.84 Gb clean data were kept.

**Illumina library construction and sequencing.** Genomic DNA was extracted from young leaves using Hi-DNAsecure Plant Kit according to the manufacturer's instructions (Tiangen Biotech, Beijing, China). Three different insert-size Illumina DNA libraries for draft assembly improvement (error correction) were constructed using the KAPA HyperPrep Kit (Kapa Biosystems, Roche, USA) following the manufacturer's recommendations. Two libraries were sequenced on an Illumina HiSeq2500 platform under paired-end 250 bp mode. One library was sequenced under paired-end 150 bp mode. A total of 205.74 Gb raw data were generated. After quality control by using fastp[42] (version 0.12.6) with default parameters, 172.52 Gb clean data were kept for draft assembly improvement.

To facilitate genome annotation, we performed mRNA sequencing of nine tissues, including leaf, spikelet, root, internode (divided into upper, middle and lower parts), rhizome, lateral bud, and seedling. Up to 400 µg total RNA per sample was purified for each sample by using TRIzol-based method (Invitrogen, CA, USA) and then treated with DNase I. The quality of extracted RNA was evaluated using 1% agarose gel. Before polyA mRNA enrichment, the integrity of RNA was checked using Aglient 2100 bioanalyzer (Agilent Technologies, Inc., CA, USA). The polyA mRNA was enriched following the protocol of NEBNext® Poly(A) mRNA Magnetic Isolation Module (New England Biolabs #7490 S, MA, USA). RNA sequencing libraries were then constructed using NEBNext® Ultra™ II RNA Library Prep Kit for Illumina (New England Biolabs #E7775, MA, USA) following the manufacturer's instructions. The concentration of RNA libraries was quantified using the Qubit® 3.0 Fluorometer. Libraries were then sent for 150 bp paired-end sequencing on an Illumina HiSeq2500 platform. In total, 95.12 Gb raw data were generated for nine tissue samples.

**Hi-C library construction and sequencing.** The same plant used for Oxford Nanopore sequencing was used for Hi-C library construction. Tender leaves were collected for Hi-C library construction with a modified method[43,44]. Briefly, young leaf sample was fixed with formaldehyde. Fixed sample was then lysed, and the cross-linked DNA was digested with restriction endonuclease DpnII (New England Biolabs, USA) overnight. Biotin-labeled bases were introduced during the sticky end repairing process. The interacting DNA fragments were ligated to form chimeric junctions. DNA fragments were purified and physically sheared to a size of 300–700 bp. DNA fragments tagged with biotin were enriched with beads and then sent to DNA sequencing library construction. The libraries were sent to be sequenced on an Illumina HiSeq4000 platform under paired-end 150 bp mode. In total, 347.76 Gb (~159× genome coverage) clean data were generated after quality control by fastp[42] (version 0.12.6) with default parameters. The quality of Hi-C data was evaluated using HiC-Pro[45] (Supplementary Tables 27 and 28).

**Estimation of genome size and heterozygosity.** The genome size of *M. lutarioriparius* was estimated using k-mer frequency distribution generated from Illumina short reads. The program jellyfish[46] (version 2.2.9) was used to calculate 17-mer frequency distribution. GCE (Genome Characteristics Estimation) software[47] (ftp://ftp.genomics.org.cn/pub/gce) was then used to estimate the genome size of *M. lutarioriparius* with modified parameters: -c 24, -m 1, -D 8 -H 1.

**Genome assembly**. Before assembly, raw Oxford Nanopore reads were self-corrected using Canu[48] (v1.8) with parameters: genomeSize = 2 G, minReadLength = 2000, minOverlapLength = 500, -nanopore-corrected. After self-correction, a total of 112.78 Gb data (the longest 40× reads) were kept. The corrected Nanopore reads were then assembled with SMARTdenovo (https://github.com/ruanjue/smartdenovo) with parameters: -k 21, the others were set as default. The initial contig assembly was polished through three iterations using Racon (https://github.com/lbcb-sci/racon) with raw Nanopore reads and default parameters. Contigs were further polished with Pilon[49] (v1.23) through three iterations using 172.52 Gb filtered Illumina data. The Pilon parameters are: -fix bases, -changes, -vcf, -diploid. The quality of assembly after each round of polish was assessed using BUSCO.

**Hi-C scaffolding**. To confirm the reliability of the Hi-C scaffolding, two Hi-C scaffolding programs LACHESIS[14] (http://shendurelab.github.io/LACHESIS/) and 3d-dna pipeline[15] (https://github.com/theaidenlab/3d-dna) were tested to anchor the contigs in this study, and the better one that generated by LACHESIS was selected as the final assembly.

Before scaffolding, raw Hi-C reads were removed of adapters and trimmed for low-quality bases using fastp[42] (version 0.12.6) with default parameters. A total of 4772 M clean reads were mapped to the contig assembly using BWA[50] (version 0.7.17). The deduplicated list of alignments of Hi-C reads to the contig assembly was generated using Juicer pipeline[51] (version 1.5.7). Then, 3d-dna[15] (version 180922) was used to anchor the contig assembly into chromosome-length scaffolds. 3d-dna was tested to run with haploid and diploid mode, respectively. The iterative rounds for mis-correction were tested with 0, 2, and 7 times, respectively. The heatmap for Hi-C interaction was processed using 3d-dna visualize module and reviewed in Juicebox[52] (version 1.9.0).

LACHESIS was also used to construct chromosome-level assembly. The quality filtered Hi-C reads were mapped to the Nanopore contigs assembly using the BWA-aln algorithm[50] (version 0.7.17). For assembly correction, contigs were split into 50 kb fragments then Hi-C reads were used to recover the contigs. The duplicated reads were removed using MarkDuplicates of Picard. And then the PreprocessSAMs.pl program was used to further filter the alignment files. Finally, the LACHESIS was used to anchor the Nanopore contigs into pseudochromosomes with parameters: CLUSTER_N = 19; CLUSTER_MIN_RE_SITES = 22; CLUSTER_MAX_LINK_DENSITY = 2; CLUSTER_NONINFORMATIVE_RATIO = 2; ORDER_MIN_N_RES_IN_TRUN = 10; ORDER_MIN_N_RES_IN_SHREDS = 10; RE_SITE_SQR = GATC in the configure file. The 19 pseudochromosomes of *M. lutarioriparius* constructed by Hi-C technologies were named Chr01 to Chr19 according to the syntenic relationship with sorghum.

**Genome assembly quality evaluation**. The quality of Nanopore contigs was assessed by sequence comparison with finished BAC sequences, which were sequenced using Sanger sequencing. Sequence comparisons between Nanopore contigs and BAC sequences were performed using MUMmer[53] (version 3.23). Reads coverage for contigs was calculated by mapping raw Nanopore reads against BAC sequences using minimap2[54] (version 2.16) with parameters: -x map-ont -a. BUSCO[55] (version: 3.1.0, database: embryophyta_odb10) was used to assess the completeness of genome assembly and predicted protein-coding genes. Three Illumina DNA sequencing libraries were mapped to the genome assembly using BWA-MEM[50] (version 0.7.17), and the mapping rates were calculated. Nine RNA sequencing libraries were mapped against the genome assembly using HISAT2[56] (version 2.0.5), and the overall alignment rates were calculated. LTR Assembly Index (LAI)[16] was used to evaluate the assembly quality. LTR_retriever[57] was used to accurately identify LTR-RTs and calculate the LAI using the output of LTR-FINDER[58].

**Repeat analysis**. A repeats library of *M. lutarioriparius* genome was ab initio constructed using RepeatModeler (open-1.0.11) (http://www.repeatmasker.org/RepeatModeler). The consensus TE sequences generated by RepeatModeler software were combined with RepBase-20181026 and used as repeats library in RepeatMasker (version open-4.0.7) (http://www.repeatmasker.org) for repetitive elements identification in *M. lutarioriparius* genome. A preliminary list of candidate LTR-RT was generated using LTR_FINDER[58] (version 1.0.7) with parameters: -D 20000 -d 1000 -L 5000 -l 100 -p 20 -C. The identification of high-quality intact LTR-RTs and the calculation of insertion age for intact LTR-RTs were carried out using LTR_retriever[57] with default parameters. De novo searches for MITEs used MITE Tracker software[59] with default parameters. Tandem repeats detection used Tandem Repeats Finder program[60] (version 4.09) with parameters: 2 7 7 80 10 50 500 -m -f -d. The locations of centromere and telomere were inferred from the outputs generated by Tandem Repeats Finder.

**RNA sequencing data analysis**. The RNA sequencing reads were removed of adapters and trimmed for low-quality bases using fastp[42] (version 0.12.6) with parameters: -q 25 -l 75. After quality control, 82.34 Gb clean data were kept for gene prediction and expression analysis. Clean reads were mapped to the *M. lutarioriparius* genome using HISAT2[56] (version 2.0.5) with default parameters. Following the alignment, SummarizedExperiment object was generated using

R/Bioconduct package tximeta (version 1.4.2) and raw reads count matrix for each gene were derived from SummarizedExperiment object using R function counts.

**Gene prediction and genome annotation**. Before gene prediction, the repetitive sequences were masked by RepeatMasker (http://www.repeatmasker.org/RepeatModeler.html). (1) Ab initio prediction: Fgenesh was used for gene prediction with authorized monocots model and modified parameters: -pmrna -scip_prom -scip_term -min_f_exon:16 -min_i_exon:16 -min_t_exon:16 -min_s_exon:90. Augustus[61] (version 3.2.3) was running with parameters: -strand=both -genemodel=partial -gff3=on -species=maize5. (2) Homolog protein-based gene prediction: the proteomes of rice (*O. sativa* v7_JGI), maize (*Z. mays* Ensembl-18), sorghum (*S. bicolor* v3.1.1), and *S. spontaneum*[28] were mapped to the genome assembly using Exonerate[62] (version 2.2.0) with Protein2Genome model. (3) RNA sequencing aided gene annotation: Trinity[63] (version 2.1.1) was used to perform de novo transcript reconstruction. Redundant transcript sequences were collapsed using CD-HIT[64] (version 4.7) with parameters: -c 0.90 -n 9 -d 0 -M 0 -T 0. For genome-guided transcript assembly, the RNA sequencing reads were mapped to repeats masked *M. lutarioriparius* genome assembly using HISAT2[56] (version 2.0.5). StringTie[65] (v1.3.2d) was used to reconstruct the transcripts. The PASA pipeline[66] (v2.2.0) was used to construct comprehensive transcripts by integrating transcripts output by de novo and genome-guided assembly.

The EVidenceModeler (EVM)[66] (version 1.1.1) was used to integrate the gene prediction results generated above. According to our understanding of the reliability of different software, we assigned score 5 to the prediction result of FGENESH, 3 to augustus, 5 to exonerate, and 10 to PASA in the configuration file of EVM.

MAKER[67] (version 2.31.10) was used to add MAKER's quality-control metrics to the EVM integration results. At least 85% of exons per gene should have evidence support from RNA-seq transcripts or homolog proteins, and these predicted genes with translated protein sequence length less than 50 amino acids would be filtered.

The protein sequences were subjected to InterProScan[68] (version 5.22-61.0), eggNOG-mapper[19,69], and KOALA (KEGG Orthology And Links Annotation)[70] for genome annotation. InterProScan was run with parameters of '-iprlookup -goterms -pa -f tsv' and GO annotation was extracted from the output of InterProScan. PlantTFDB 5.0[71] was used to predict transcription factor.

Non-coding RNAs were identified using Rfam (version 14) and Infernal (version 1.1.2). The tRNAs identification used tRNAscan-SE 2.0[72]. The rRNA identification used barrnap (https://github.com/tseemann/barrnap).

**Genomic comparisons and visualizations**. MCScan[27] (Python version) was used for pairwise synteny region search with the LSAT results of *M. lutarioriparius* versus sorghum. Dotplots for genome pairwise synteny visualization was generated using the command 'python -m jcvi.graphics.dotplot'. The synteny pattern of two compared genomes was using the command 'python -m jcvi.compara.synteny depth -histogram'. Microsynteny visualization was using the command 'python -m jcvi.graphics.synteny'.

**Estimate of the recent whole-genome duplication timing in *Miscanthus lutarioriparius***. MCScanX[27] was used to perform the self-comparison of *M. lutarioriparius* protein-coding genes. The synonymous substitution values ($Ks$) of *M. lutarioriparius* versus *M. lutarioriparius* syntenic gene pairs were calculated using KaKs_calculator[73] (version 2.0) with YN model. The timing of *M. lutarioriparius*' recent WGD was calculated based on the small $Ks$ peak of *M. lutarioriparius* versus *M. lutarioriparius* collinear gene pairs. The synonymous substitutions rate per site per year ($r$) equaling 6.5e-9 was applied to the estimation of recent WGD.

The genes of *M. lutarioriparius* were classified as singletons, dispersed duplicates, proximal duplicates, tandem duplicates, and segmental/WGD duplicates using duplicate_gene_classifier module in MCScanX[27] by parsing the all-vs-all BLASTP results. For the definition of proximal duplicates, two genes are in nearby genomic region and separated by less than 10 genes.

KEGG and GO enrichment analysis of gene sets were performed in R (version 3.5.0) (https://www.r-project.org) against *M. lutarioriparius* genome as reference. Statistical significance was tested using over-representation test using R package clusterProfiler[74]. Benjamini-Hochberg (BH) method was used to control the false discovery rate (under the threshold of 0.05) in multiple hypothesis testing.

**Phylogenetic analysis and divergence time estimation**. To investigate the evolutionary history of *M. lutarioriparius*, seven grass subfamilies, *Brachypodium distachyon* v3.1 (Phytozome v12.1), *Oryza sativa* v7_JGI (Phytozome v12.1), *Zea mays* Ensembl-18 (Phytozome v12.1), *Sorghum bicolor* v3.1.1 (Phytozome v12.1), *Setaria italica* v2.2 (Phytozome v12.1), *Saccharum spontaneum*[28], *Miscanthus lutarioriparius*, and one dicot plant *Arabidopsis thaliana* TAIR10 (Phytozome v12.1) were used for gene family construction by using OrthoFinder[75] (version 2.2.7) with default parameters. The protein sequences of 122 single-copy orthologs from 8 species were concatenated for the species tree construction. These concatenated single-copy orthologs (protein sequences) were aligned using MAFFT[76] (version: v7.158b) L-INS-i iterative refinement method. The best-fit model (JTT +

G + F) was selected by using ModelGenerator software[77] (v0.85). Maximum likelihood trees were constructed using RAxML[78] (8.2.12) with 1000 replicates bootstrap. BEAST (Bayesian Evolutionary Analysis Sampling Trees)[79] (v1.8.3) was used to estimate the species divergence times with JTT substitution model and Gamma categories equaling to 4. The calibrated Yule model and strict clock type was set. The length of chain for MCMC was set 10,000,000 and every 1000 record the parameters. The calibration points used in BESAT were obtained from the TimeTree database (http://www.timetree.org/): O. sativa versus S. bicolor (median time: 48.0 Ma), S. bicolor versus Z. mays (median time: 11.56 Ma). The visualization of species tree used FigTree (v1.4.3) (https://github.com/rambaut/figtree/).

**Genetic diversity and phylogeny analysis based on transcriptomes.** Raw transcriptome data were downloaded from NCBI sequence read archive (Project ID: SRP066219 [https://www.ncbi.nlm.nih.gov/bioproject/PRJNA301483]). fastp[42] (version 0.12.6) was used to filter the reads with parameter -q 25. Identification of genetic variations was performed following the workflow of RNAseq short variants discovery designed by GATK (https://github.com/gatk-workflows/gatk4-rnaseq-germline-snps-indels). Genome-wide Complex Trait Analysis (GCTA)[80] (version 1.26.0) was used to perform the principal components analysis. VCF2Dis (version 1.09) (https://github.com/BGI-shenzhen/VCF2Dis) was used to calculate p-distance. Phylip[81] (version 3.697) was used to build the neighbor-joining tree. A total of 6659 SNPs were kept (genotype missing rate <10% and minimum allele frequency >5%) and used for admixture analysis[82].

**Gene family expansion and contraction analysis.** The gene-family expansion and contraction in M. lutarioriparius genome were determined by comparing with those in 7 other species using Computational Analysis of Gene Family Evolution (CAFE)[83] (version 4.2.1). The gene family size for each species used in CAFE was calculated by OrthoFinder[75] (version 2.2.7). The ultra-metric phylogeny tree was reconstructed using r8s[84] (v1.81). We suspected that the clade of M. lutarioriparius, S. spontaneum, S. bicolor share the same rate of gene family evolution (birth and death) and rest of the species had another rate. To better understand the potential function category of each gene family, we used KinFin[85] (v1.0.3) to take gene families identified by OrthoFinder, alongside gene functional annotation assigned by InterProScan, to derive rich aggregative annotation for gene families.

**Chloroplast genome assembly and annotation.** Illumina short reads generated from whole-genome sequencing were used for chloroplast genome assembly using MITObim software[86] (version 1.9) with default parameters. Two bait sequences (Miscanthus junceus (LN869216.1) and Saccharum spontaneum (LN896360.1)) were used to verify whether different bait chloroplast genomes could affect the assembly results. The GeSeq[87] (https://chlorobox.mpimp-golm.mpg.de/geseq.html) was used to annotate protein-coding genes, rRNA genes, and tRNA genes, in the M. lutarioriparius chloroplast genome.

**Phylogeny analysis of genus Miscanthus based on chloroplast genome sequence.** We downloaded 33 Miscanthus chloroplast genomes from NCBI database (Supplementary Data 4). Four chloroplast genomes were used as outgroup, including a S. spontaneum (LN896360.1), a maize (NC001666.2), a rice (NC_031333.1), and a B. distachyon (NC_011032.1). All chloroplast genome sequences were aligned using MAFFT[76] (version: v7.158b) with -auto option. Conserved blocks of the alignment were selected using GBLOCKS[88] (version 0.91b). IQ-TREE[89] (version 1.6.3) was used to reconstruct the maximum likelihood phylogenetic tree with 1000 replicates of ultrafast bootstrap. Bayesian inference (BI) of phylogeny was carried out using MrBayes[90] (version 3.1.2). The number of generations was set to 10,000,000, and the sample frequency of Markov chain was set to 100 generations; Nst was set to 6 and rates was set to Gamma during the execution of MrBayes; S. spontaneum (LN896360.1) was set as outgroup; 25% of the samples were discard as burn-in. For neighbor-joining phylogenetic tree reconstruction, MEGA X[91] was used with 1000 bootstrap trails. Time-calibrated phylogeny analysis based on the chloroplast genome sequences was carried out using BEAST (Bayesian Evolutionary Analysis Sampling Trees) software[79] (v1.8.3). Sites model was set as HKY, and site heterogeneity model was set as Gamma + Invariant sites. Number of gamma categories was set as 4. Length of chain was set as 10,000,000. Log parameters every 1000. Only one calibrations point from TimeTree (http://www.timetree.org/) was used (Saccharum spontaneum versus Miscanthus lutarioriparius, median time: 6.98 Ma, range: 5.53–8.43 Ma).

**Identification of plant disease resistance genes.** We used DRAGO 2 tool of Plant Resistance Genes database (PRGdb 3.0, http://prgdb.org/prgdb/genes)[92] to predict candidate Pathogen Receptor Genes (PRGs). The R genes of M. lutarioriparius genome were further classified into CNL (CC-NB-LRR), TNL (TIR-NB-LRR), RLP (ser/thr-LRR), RLK (Kin-LRR), and others based on the presence of specific domains.

We also employed HMMER software[93] (version 3.1 b1) to identify NBS-LRR genes in M. lutarioriparius using Hidden Markov Model (HMM) profile for Pfam[94] NBS domain (NB-ARC, PF00931, http://pfam.xfam.org/). A high-quality protein set (E-value < $1 \times 10^{-4}$) was kept as candidate for further filtration. In addition, 231

reference sequences of NBS-LRR genes were download from NCBI and used for BLASTP[95] (version 2.2.28+) search. The E-value for BLAST was set as 0.01.

Only these genes identified by both DRAGO 2 tool and HMMER software would be taken as candidate NBS-LRR genes. The proteins of candidate NBS-LRR genes were then subjected to Batch Web CD-Search (https://www.ncbi.nlm.nih.gov/Structure/) for function domain annotation. The final candidate NBS-LRR genes were filtered by manual check according to the results of Batch Web CD-Search and BLASTP. The NBS-associated conserved domains TIR (PF01582) and RPW8 (PF05659) were identified using HMMER software with default parameters.

**Identification and phylogeny analysis of cell-wall-biosynthesis-related gene families in M. lutarioriparius.** Known cellulose synthases (CesA) and cellulose synthase-like (Csl) CDSs of rice, maize, and Arabidopsis were downloaded from the Cell Wall genomics (https://cellwall.genomics.purdue.edu/families/). We performed BLAST[95] (version 2.2.28+) search against the CDSs of M. lutarioriparius with these cellulose synthases and cellulose synthase-like protein sequences. BLAST E-value threshold was set to 1e-5. The BLAST hints of CesA were manually filtered based on that the cumulative length of same target gene should not be less than 1300 bp. HMM profiles of CesA (PF03552) and Csl (PF00535) obtained from Pfam database[94] (http://pfam.xfam.org) were used to search against M. lutarioriparius protein sequences using HMMER[93] with cut_tc mode. The E-value cutoff was set to 1e-3. We kept the results detected by both HMMER and BLAST search, which means every candidate protein sequence should be similar to the BLAST query and own either of the two Pfam domains.

Using 12 sorghum CesA genes annotated by Phytozome v12.1 (https://phytozome.jgi.doe.gov/), 28 candidate CesA genes were identified in M. lutarioriparius based on the genome collinearity. One of these 28 candidate genes was removed due to the CDS length (Ml02G069810: 429 bp). Rest of the 27 genes were well supported by both HMMER and BLAST search above.

The CesA and Csl protein sequences were aligned using MAFFT[76] (version: v7.158b) with L-INS-i method. The maximum likelihood tree was reconstructed using IQ-TREE[89] (version 1.6.3) with 1000 replicates of ultrafast bootstrap. Gene structure was displayed using Gene Structure Display Server (GSDS 2.0) (http://gsds.cbi.pku.edu.cn).

dbCAN2 meta server (http://bcb.unl.edu/dbCAN2/) was used to annotate carbohydrate-active enzyme (CAZyme) in M. lutarioriparius. dbCAN2 meta server integrates three tools HMMER, DIAMOND, and Hotpep (DIAMOND: E-value < 1e-102, hits per query (-k) = 1; HMMER: E-value < 1e-15, coverage > 0.35; Hotpep: frequency > 2.6, hits > 6). CAZymes are defined as those predicted by at least two tools.

**Reporting summary.** Further information on research design is available in the Nature Research Reporting Summary linked to this article.

## Data availability

Data supporting the findings of this work are available within the paper and its Supplementary Information files. The datasets and plant material generated and analyzed during the current study are available from the corresponding author upon request. A reporting summary is available as a Supplementary Information file. Oxford Nanopore whole-genome sequencing reads, Illumina genomic (including Hi-C experiment) and transcriptomic reads are available from the EBI database with study accession code PRJEB40463. The genome assembly, annotations, chloroplast genome assembly and other data are available from figshare [https://figshare.com/projects/Miscanthus_lutarioriparius_genome_sequencing_assembly_and_annotation/89648]. Source data are provided with this paper.

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

## Acknowledgements

We thank Qing Gao and Jinxing Chen from CAS Center of Excellence in Molecular Plant Sciences, Institute of Plant Physiology and Ecology for greenhouse management. We thank Ying Lu from College of Fisheries and Life Science, Shanghai Ocean University for valuable advice on genome annotation. We thank Zhen Li, Danfeng Lyv, Minhuan Zeng, Lin Weng, and Li Liu from CAS Center of Excellence in Molecular Plant Sciences, Institute of Plant Physiology and Ecology for valuable suggestions. This work was supported by grant from the Chinese Academy of Sciences (XDB27010301 and KSCX2-YW-G-034 to B.H.) and the National Natural Science Foundation of China (31788103).

## Author contributions

B.H. and T.S. conceived the project and its components. J.M., Q.F., and B.H. designed the studies and experiments. Q.F., J.Y., and J.M. contributed to the collection and cultivation of *Miscanthus* samples. J.Y. provided the genome size estimation by flow cytometry. C.Z., D.F., Yiqi Lu, Q.T., W.L., Q.W., and Q.F. performed DNA and RNA preparation for genome and transcriptome sequencing. J.M., H.L., and Q.Z. performed de novo genome assembly. L.Z. performed BAC sequence assembly and sequencing data preprocessing. J.M. and Y.L. performed genome data analysis and transcriptome analyses. Y.Z. and T.H. provided IT support. L.L. and X.H. supervised the project. J.M., Q.F., and B.H. analyzed the whole data and wrote the manuscript.

## Competing interests

The authors declare no competing interests.
