## [Peer Review File · Nature Communications]

REVIEWER COMMENTS

Reviewer #1 (Remarks to the Author):

This manuscript by Miao et al. reports a chromosome-scale genome assembly for *Miscanthus lutarioriparius* (Mlu). *Miscanthus* is a broadly distributed grass with interesting traits and genomic biology, and is less studied compared to its other close relatives. Thus, this genome sequence will be of high scientific interest, valuable for both further genetic improvement and the insights it may offer into plant evolution.

This is a high-quality chromosome-scale assembly for a complex genome, known to harbor a recent whole-genome duplication and a high level of heterozygosity as an obligate outcrossing species. The assembly combines data from Oxford Nanopore long reads and Hi-C data for scaffolding, plus three Illumina pair-end libraries with different insert sizes. The assembly statistics show high coverage of the expected *Miscanthus* genome, large contigs sizes, and a high mapping rate of RNASeq data. Aside from computational assessments of assembly accuracy, the verification of the assembly was limited to Sanger sequencing of 3 BACs, and extensive macrosynteny to sorghum as expected from prior genetic mapping.

Fig 2a showed that the genome assembly contains a number of inter-chromosomal exchanges predicted to be associated with the whole genome duplication, but not observed in other genetic maps constructed from *Miscanthus sinensis* (Msi) and *Miscanthus sacchariflorus* (Msa) parents (e.g. Dong et al., 2017, GCB Bioenergy). These discrepancies raise the issue of assembly errors, but could reflect structural variations that distinguish Mlu from Msa or Msi. The authors should independently verify these computationally predicted variations by in silico positioning of SNPs from available genetic maps. As a complementary evolutionary analysis, markers selected to track these chromosomal exchanges could be explored for degree of association in other accessions within the *Miscanthus* genus. Curiously, the authors also do not compare their Mlu assembly to the chromosome-scale assembly for its closest relative Msi, which was generated by a large international consortium and has been publicly available at Phytozome since 2017. A recent inspection of Phytozome shows the Mlu and Msi genome assemblies are of similar size and number of predicted gene models. The authors should consider this genome in their comparisons.

Throughout the manuscript, the authors emphasize that Mlu is a distinct species, whereas molecular diversity data reported previously (e.g. Clark et al. 2019, Annals of Botany) instead indicate Mlu is a type of Msa. The results from chloroplast DNA phylogenies presented in Figure 4 affirms this interpretation, as Mlu clusters with tetraploid Msa and the triploid Mxg that contains two Msa and one Msi genomes. The methods provide no details about the Mlu plant that was sequenced other than it was collected from Honghu city. The authors should provide more details on the sequenced plant (akin to a voucher that would be deposited in a germplasm collection). In addition, they must directly address the prior interpretations that Mlu is a differentiated subpopulation within Msa, rather than a distinct species, or provide evidence for why it should be considered a distinct species.

Another major claim of the manuscript is that the Mlu genome harbors a significantly larger proportion of tandemly duplicated sequences compared to seven other plant genomes, which may be associated with evolution and adaptation of *Miscanthus*. This interpretation is supported by the predicted functions enriched among tandemly duplicated genes, which includes cell wall synthesis, photosynthesis, stress responses, and pollen response, traits that might enhance fitness in a broadly adapted perennial with a self-incompatible breeding system. The implication from Figure 3 is that higher gene family members is a response (retention) to phenotypic selection and adaptation. However, one caution to the gene family analysis is that among the 7 plant species compared for Figure 3, Mlu is the only one of these to have experienced a recent likely allopolyploidy event, (Ssp is autopolyploid, Zm diploidized ancient tetraploid, others all diploids). Thus, the gene family size may be inflated in Mlu because the whole genome duplication has not yet progressed far along the path to

diploidization compared to the other species. The observation that many of these duplicated genes are not highly expressed (a sign of pseudogenization), raises the possibility that they may not be major contributors to plant function. Perhaps these interpretations could be strengthened by an evolutionary analysis of some of these genes (e.g the CA genes) among a broader diversity of *Miscanthus*?

Minor Edits:

The manuscript was well-written overall. I noted two places of awkward wording at the beginning and suggest edits to improve clarity:

1. Lines 37-39 of the Abstract, the meaning here is not clear. Most of the gene duplicates are not expressed, so then probably not functional? Are these CNV/PAVs?
2. Line 46, "the chilling even cold, which made" should be "chilling, even cold, which makes"

Reviewer #2 (Remarks to the Author):

In the manuscript entitled "Chromosome-scale assembly and analysis of biomass crop *Miscanthus lutarioriparius* genome", Miao and colleagues assembled chromosome-scale *Miscanthus lutarioriparius* genome based on Nanopore sequences and HiC, explored the genome structures, analyzed several key gene families associate with biomass and estimated the divergence of *Miscanthus*. As I known, the quality of current genome is much improved assembly than the available *Miscanthus* genome so far in despite of the potential redundancy assembly. However, in comparison with the recent published genome projects, the study only conducted very basic analysis for the genome based on de novo sequences and limited RNA-seq. In addition, *Miscanthus* genome has already be available. Thus, the study only can provide very limited novel discovery for genome evolution and the genetic basis of biomass in *Miscanthus lutarioriparius*. And I don't think the current manuscript meet standard requirements for Nature Communication.

Some general comments/suggestions on the analyses conducted:

1. Line 190-192 "2,802 (8.21%) sorghum genes have more than two syntenic genes in *M. lutarioriparius* genome, which presumably resulted from the segmental duplication, tandem duplication or single gene duplication occurred in *M. lutarioriparius* genome after it split with sorghum." This phenomenon is usually caused by the heterozygosity of the genome since *Miscanthus* is self-incompatible. Hereinafter, the authors had the conclusions that "the heterozygosity in *Miscanthus* genome hindered the advances of the genome sequencing and assembly in the past". The authors may try to use the relative bioinformatic tools (for example: `purge_haplotigs` (https://bitbucket.org/mroachawri/purge_haplotigs/src/master/) to remove the potential heterozygosity assembly and then manually check the reassembled genome.

2. Line 204-207: "Besides, inter-chromosomal exchanges between M1Chr09/10 and M1Chr14/15 were observed, which probably occurred prior to the *M. lutarioriparius* recent WGD (indicated with green circles in Fig. 2a)." The tip of two chromosomes shared high collinearly in major Poaceae lineages, which is caused by concerted evolution of a homoeologous chromosome pair. The authors should review the relative publications---Paterson, A. H. et al. *Nature* 457, 551 (2009). Wang, X., Tang, H. & Paterson, A.. *Plant Cell* 23, 27-37 (2011). Wang, X et al, have estimated that the gene conversions were occurred ~13Mya in sorghum. The authors may estimate the divergence time by themselves based on the Ks of homologous gene pairs.

3. C4 pathway genes of *M. lutarioriparius* could be an interesting topic, I did not see any evidence to support the C4 characteristics (probably the gene expression pattern of leave?) of these genes beside

the orthologues analysis. Since the WGD were occurred in *M. lutarioriparius* ~ 6 MYA, two copies derived from the WGD may be divergence for their functions including C4 characteristics. The author may refer to Li, P., (2010. *Nature Genetics* 42, 1060) and Pick,T.R.et al.(2011, *Plant Cell* 23, 4208.) for the identification of C4 genes. .

Some minor issues:

1. Fig. 1a. Why the chromosome ID in both the X-axis and Y-axis were not based on the order? It is probably accorded to the length of pseudo chromosomes?

The legend for this figure --"The green circles indicate the inter-chromosomal exchanges among the chromosome 441 ends of M1Chr09, M1Chr10, M1Chr14 and M1Chr15, which probably happened before the *M. lutarioriparius* specific WGD." is not appropriate. Please see my comment above.

2. "The genome size estimated by k-mer statistics is about 81 2.19 Gb, close to 82 2.147 Gb determined using flow cytometry". The significant figures should be consistent.

3. Line: 123: 99,78% should be 99.78%

4. Some of figures in Supplementary are readable, for examples: Supplementary Fig. 26 , Supplementary Fig. 32a,

Reviewer #3 (Remarks to the Author):

This is an excellent investigation, executed with the appropriate techniques and presented in a clear and concise manner. All of the methods were appropriate, and the results have been thoroughly and carefully evaluated. This project provides valuable new information that will be useful for plant genomics research, especially in areas concerning the biology of grasses.

The only criticism I have of the manuscript is that it needs to be carefully edited for improved wording. The organization and content is excellent, but the writing has numerous grammatical errors. These should be relatively easy to correct. I have indicated a substantial number of revisions on the scanned copy included in this review, but recommend that the manuscript be carefully revised for grammar.

Reviewers' comments:

Reviewer #1 (Remarks to the Author):

This manuscript by Miao et al. reports a chromosome-scale genome assembly for *Miscanthus lutarioriparius* (Mlu). *Miscanthus* is a broadly distributed grass with interesting traits and genomic biology, and is less studied compared to its other close relatives. Thus, this genome sequence will be of high scientific interest, valuable for both further genetic improvement and the insights it may offer into plant evolution.

This is a high-quality chromosome-scale assembly for a complex genome, known to harbor a recent whole-genome duplication and a high level of heterozygosity as an obligate outcrossing species. The assembly combines data from Oxford Nanopore long reads and Hi-C data for scaffolding, plus three Illumina pair-end libraries with different insert sizes. The assembly statistics show high coverage of the expected *Miscanthus* genome, large contigs sizes, and a high mapping rate of RNASeq data. Aside from computational assessments of assembly accuracy, the verification of the assembly was limited to Sanger sequencing of 3 BACs, and extensive macrosynteny to sorghum as expected from prior genetic mapping.

Fig 2a showed that the genome assembly contains a number of inter-chromosomal exchanges predicted to be associated with the whole genome duplication, but not observed in other genetic maps constructed from *Miscanthus sinensis* (Msi) and *Miscanthus sacchariflorus* (Msa) parents (e.g. Dong et al., 2017, GCB Bioenergy). These discrepancies raise the issue of assembly errors, but could reflect structural variations that distinguish Mlu from Msa or Msi. The authors should independently verify these computationally predicted variations by in silico positioning of SNPs from available genetic maps. As a complementary evolutionary analysis, markers selected to track these chromosomal exchanges could be explored for degree of association in other accessions within the *Miscanthus* genus. Curiously, the authors also do not compare their Mlu assembly to the chromosome-scale assembly for its closest relative Msi, which was generated by a large international consortium and has been publicly available at Phytozome since 2017. A recent inspection of Phytozome shows the Mlu and Msi genome assemblies are of similar size and number of predicted gene models. The authors should consider this genome in their comparisons.

Throughout the manuscript, the authors emphasize that Mlu is a distinct species, whereas molecular diversity data reported previously (e.g. Clark et al. 2019, Annals of Botany) instead indicate Mlu is a type of Msa. The results from chloroplast DNA phylogenies presented in Figure 4 affirms this interpretation, as Mlu clusters with tetraploid Msa and the triploid Mxg that contains two Msa and one Msi genomes. The methods provide no details about the Mlu plant that was sequenced other than it was collected from Honghu city. The authors should provide more details on the sequenced plant (akin to a voucher that would be deposited in a germplasm collection). In addition, they must directly address the prior interpretations that Mlu is a differentiated subpopulation within Msa, rather than a distinct species, or provide evidence for why it should be considered a distinct species.

Another major claim of the manuscript is that the Mlu genome harbors a significantly larger proportion of tandemly duplicated sequences compared to seven other plant genomes, which may be associated with evolution and adaptation of *Miscanthus*. This interpretation is supported by the predicted functions enriched among tandemly duplicated genes, which includes cell wall synthesis, photosynthesis, stress responses, and pollen response, traits that might enhance fitness in a broadly adapted perennial with a self-incompatible breeding system. The implication from Figure 3 is that higher gene family members is a response (retention) to phenotypic selection and adaptation.

However, one caution to the gene family analysis is that among the 7 plant species compared for Figure 3, Mlu is the only one of these to have experienced a recent likely allopolyploidy event, (Ssp is autopolyploid, Zm diploidized ancient tetraploid, others all diploids). Thus, the gene family size may be inflated in Mlu because the whole genome duplication has not yet progressed far along the path to diploidization compared to the other species. The observation that many of these duplicated genes are not highly expressed (a sign of pseudogenization), raises the possibility that they may not be major contributors to plant function. Perhaps these interpretations could be strengthened by an evolutionary analysis of some of these genes (e.g the CA genes) among a broader diversity of *Miscanthus*?

Minor Edits:

The manuscript was well-written overall. I noted two places of awkward wording at the beginning and suggest edits to improve clarity: 1. Lines 37-39 of the Abstract, the meaning here is not clear. Most of the gene duplicates are not expressed, so then probably not functional? Are these CNV/PAVs?

2. Line 46, “the chilling even cold, which made” should be “chilling, even cold, which makes”

Reviewer #2 (Remarks to the Author):

In the manuscript entitled “Chromosome-scale assembly and analysis of biomass crop *Miscanthus lutarioriparius* genome”, Miao and colleagues assembled chromosome-scale *Miscanthus lutarioriparius* genome based on Nanopore sequences and Hi-C, explored the genome structures, analyzed several key gene families associate with biomass and estimated the divergence of *Miscanthus*. As I known, the quality of current genome is much improved assembly than the available *Miscanthus* genome so far in despite of the potential redundancy assembly. However, in comparison with the recent published genome projects, the study only conducted very basic analysis for the genome based on de novo sequences and limited RNA-seq. In addition, *Miscanthus* genome has already be available. Thus, the study only can provide very limited novel discovery for genome evolution and the genetic basis of biomass in *Miscanthus lutarioriparius*. And I don't think the current manuscript meet standard requirements for Nature Communication.

Some general comments/suggestions on the analyses conducted:

1. Line 190-192 “2,802 (8.21%) sorghum genes have more than two syntenic genes in *M. lutarioriparius* genome, which presumably resulted from the segmental duplication, tandem duplication or single gene duplication occurred in *M. lutarioriparius* genome after it split with sorghum.” This phenomenon is usually caused by the heterozygosity of the genome since *Miscanthus* is self-incompatible. Hereinafter, the authors had the conclusions that” the heterozygosity in *Miscanthus* genome hindered the advances of the genome sequencing and assembly in the past”. The authors may try to use the relative bioinformatic tools (for example: `purge_haplotigs` (https://bitbucket.org/mroachawri/purge_haplotigs/src/master/) to remove the potential heterozygosity assembly and then manually check the reassembled genome.

2. Line 204-207: “Besides, inter-chromosomal exchanges between M1Chr09/10 and M1Chr14/15 were observed, which probably occurred prior to the *M. lutarioriparius* recent WGD (indicated with green circles in Fig. 2a).” The

tip of two chromosomes shared high collinearity in major Poaceae lineages, which is caused by concerted evolution of a homoeologous chromosome pair. The authors should review the relative publications---Paterson, A. H. et al. Nature 457, 551 (2009). Wang, X., Tang, H. & Paterson, A.. Plant Cell 23, 27–37 (2011). Wang, X et al, have estimated that the gene conversions were occurred ~13Mya in sorghum. The authors may estimate the divergence time by themselves based on the Ks of homologous gene pairs.

3. C4 pathway genes of *M. lutarioriparius* could be an interesting topic, I did not see any evidence to support the C4 characteristics (probably the gene expression pattern of leaf?) of these genes beside the orthologues analysis. Since the WGD were occurred in *M. lutarioriparius* ~ 6 MYA, two copies derived from the WGD may be divergence for their functions including C4 characteristics. The author may refer to Li, P., (2010. Nature Genetics 42, 1060) and Pick, T.R. et al. (2011, Plant Cell 23, 4208.) for the identification of C4 genes.

Some minor issues:

1. Fig. 1a. Why the chromosome ID in both the X-axis and Y-axis were not based on the order? It is probably accorded to the length of pseudo chromosomes?

The legend for this figure –"The green circles indicate the inter-chromosomal exchanges among the chromosome 441 ends of MIChr09, MIChr10, MIChr14 and MIChr15, which probably happened before the *M. lutarioriparius* specific WGD." is not appropriate. Please see my comment above.

2. "The genome size estimated by k-mer statistics is about 81 2.19 Gb, close to 82 2.147 Gb determined using flow cytometry". The significant figures should be consistent.

3. Line: 123: 99,78% should be 99.78%.

4. Some of figures in Supplementary are readable, for examples: Supplementary Fig. 26, Supplementary Fig. 32a,

Reviewer #3 (Remarks to the Author):

This is an excellent investigation, executed with the appropriate techniques and presented in a clear and concise manner. All of the methods were appropriate, and the results have been thoroughly and carefully evaluated. This project provides valuable new information that will be useful for plant genomics research, especially in areas concerning the biology of grasses.

The only criticism I have of the manuscript is that it needs to be carefully edited for improved wording. The organization and content are excellent, but the writing has numerous grammatical errors. These should be relatively easy to correct. I have indicated a substantial number of revisions on the scanned copy included in this review, but recommend that the manuscript be carefully revised for grammar.

Detailed Response to Reviewers

Reviewer #1 (Remarks to the Author):

This manuscript by Miao et al. reports a chromosome-scale genome assembly for *Miscanthus lutarioriparius* (Mlu). *Miscanthus* is a broadly distributed grass with interesting traits and genomic biology and is less studied compared to its other close relatives. Thus, this genome sequence will be of high scientific interest, valuable for both further genetic improvement and the insights it may offer into plant evolution.

This is a high-quality chromosome-scale assembly for a complex genome, known to harbor a recent whole-genome duplication and a high level of heterozygosity as an obligate outcrossing species. The assembly combines data from Oxford Nanopore long reads and Hi-C data for scaffolding, plus three Illumina pair-end libraries with different insert sizes. The assembly statistics show high coverage of the expected *Miscanthus* genome, large contigs sizes, and a high mapping rate of RNASeq data. Aside from computational assessments of assembly accuracy, the verification of the assembly was limited to Sanger sequencing of 3 BACs, and extensive macrosynteny to sorghum as expected from prior genetic mapping.

Thanks for the comments.

Fig 2a showed that the genome assembly contains a number of inter-chromosomal exchanges predicted to be associated with the whole genome duplication, but not observed in other genetic maps constructed from *Miscanthus sinensis* (Msi) and *Miscanthus sacchariflorus* (Msa) parents (e.g. Dong et al., 2017, GCB Bioenergy). These discrepancies raise the issue of assembly errors but could reflect structural variations that distinguish Mlu from Msa or Msi. The authors should independently verify these computationally predicted variations by in silico positioning of SNPs from available genetic maps. As a complementary evolutionary analysis, markers selected to track these chromosomal exchanges could be explored for degree of association in other accessions within the *Miscanthus* genus.

Curiously, the authors also do not compare their Mlu assembly to the chromosome-scale assembly for its closest relative Msi, which was generated by a large international consortium and has been publicly available at Phytozome since 2017. A recent inspection of Phytozome shows the Mlu and Msi genome assemblies are of similar size and number of predicted gene models. The authors should consider this genome in their comparisons.

Thanks for your suggestions. Before we submitted our manuscript, the paper of DOE-JGI Phytozome *Miscanthus sinensis* genome had not been published. As the statement of DOE-JGI regarding to the *Miscanthus sinensis* genome data, DOE-JGI did not allow the use (for publication) of *Miscanthus sinensis* genome for comparative genomics analysis prior to publication by JGI and/or its collaborators. That was the reason why we did not compare our *Miscanthus lutarioriparius* to the chromosome-scale assembly of *Miscanthus sinensis* in our manuscript. Since the paper of *Miscanthus sinensis* has been published recently, so we can perform the comparative genomics analysis for *Miscanthus sinensis* and our *Miscanthus lutarioriparius*.

By performing gene synteny analysis between *Miscanthus sinensis* and *Sorghum bicolor*, we also found the dots that indicated by green circles in figure 2a of our manuscript, indicating this feature is also shared by *Miscanthus sinensis* (Fig.1). According to the suggestion from reviewer #2, we know that the rice chromosomes 11 and 12, sorghum chromosomes 5 and 8 (homologues chromosomes of M1Chr09/10 and M1Chr14/15) both share an ~3 Mb duplicated DNA segment at the termini of their short arms (Paterson et al.,

2009; Wang et al., 2011). So, we changed our description "inter-chromosomal exchanges between M1Chr09/10 and M1Chr14/15 were observed, which probably occurred prior to the *M. lutarioriparius* recent WGD (indicated with green circles in Fig. 2a)" in our manuscript with following: The ends of M1Chr09/10 and M1Chr14/15 are highly collinear, which common to major Poaceae lineages (Paterson et al., 2009; Wang et al., 2011) (indicated with green circles in Fig. 3a). Therefore, we are confident that this is not an assembly error or structural variations that distinguish Mlu from Msa or Msi.

Fig. 1 Gene synteny analysis for *Miscanthus sinensis* and *Sorghum bicolor*.

We also compared our *Miscanthus lutarioriparius* assembly with recently published *Miscanthus sinensis* assembly. There is a high collinearity between *Miscanthus lutarioriparius* and *Miscanthus sinensis* (Fig. 2). The sequence similarity between *M. lutarioriparius* and *M. sinensis* is about 47.67% (Supplementary Table 26). A large number of small inversions were observed between two genome assemblies, which may be due to assembly error or species differences (Fig.2a of this response letter). In our revised manuscript, we added a new section named "Centromeric evolution in *Miscanthus lutarioriparius*", which confirmed the allotetraploid origin of *M. lutarioriparius* using the centromeric satellite repeats.

Furtherly, we assigned each *M. lutarioriparius* chromosome in bulk to the A and B subgenome through calculating the synonymous substitution rate (K_s) for homologues chromosomes of *M. lutarioriparius* and *M. sinensis* (Fig. 2b). And we added this part of content in the 'Discussion' section of our revised manuscript.

Fig. 2 Gene synteny analysis for *Miscanthus lutarioriparius* and *Miscanthus sinensis*.

(a) Gene synteny analysis for *Miscanthus lutarioriparius* and *Miscanthus sinensis*. (b) Violin plot of synonymous substitution rates of syntenic gene pairs of homologues chromosome pairs of *Miscanthus lutarioriparius* and *Miscanthus sinensis*. The right panel is the barplot of number of syntenic gene pairs of homologues chromosome pairs of *Miscanthus lutarioriparius* and *Miscanthus sinensis*.

Supplementary Table 26 Sequence similarity between *M. lutarioriparius* and *M. sinensis*

Sub-genome	M. sinensis	M. lutarioriparius	Gene count of M. sinensis	Gene count of M. lutarioriparius	Sequence similarity (%)
A	MsChr01	MIChr01	5,190	5,347	48.3
A	MsChr03	MIChr04	3,699	3,766	47.51
A	MsChr05	MIChr06	4,123	4,226	47.93
A	MsChr08	MIChr08	3,346	3,328	48.11
A	MsChr10	MIChr09	2,036	2,902	46.88
A	MsChr11	MIChr12	2,765	2,801	47.32
A	MsChr13	MIChr13	2,007	2,129	47.16
A	MsChr14	MIChr15	1,707	1,673	47.61
A	MsChr17	MIChr16	2,701	2,446	47.94
A	MsChr18	MIChr19	2,833	2,492	47.69
B	MsChr02	MIChr02	5,163	5,352	48.22
B	MsChr04	MIChr03	4,097	4,322	47.81
B	MsChr06	MIChr05	4,160	4,326	47.48
B	MsChr07	MIChr07	5,579	5,429	47.6
B	MsChr09	MIChr10	2,124	3,439	46.44
B	MsChr12	MIChr11	2,828	2,579	47.54
B	MsChr15	MIChr14	2,087	2,692	48.74
B	MsChr16	MIChr17	2,642	2,650	47.84
B	MsChr19	MIChr18	2,799	2,933	47.64

We also provided gene synteny analysis of *M. lutarioriparius* versus *S. spontaneum* and *M. sinensis* versus *S. spontaneum* for better understanding the genome variation between *M. lutarioriparius* and *M. sinensis* (Fig. 3).

Fig. 3 Gene synteny analysis between *Miscanthus* species and *Saccharum spontaneum*

(A) Gene synteny analysis for *Miscanthus lutarioriparius* and *Saccharum spontaneum*. (B) Gene synteny analysis for *Miscanthus sinensis* and *Saccharum spontaneum*.

Throughout the manuscript, the authors emphasize that Mlu is a distinct species, whereas molecular diversity data reported previously (e.g. Clark et al. 2019, *Annals of Botany*) instead indicate Mlu is a type of Msa. The results from chloroplast DNA phylogenies presented in Figure 4 affirms this interpretation, as Mlu clusters with tetraploid Msa and the triploid Mxg that contains two Msa and one Msi genomes. The methods provide no details about the Mlu plant that was sequenced other than it was collected from Honghu city. The authors should provide more details on the sequenced plant (akin to a voucher that would be deposited in a germplasm collection). In addition, they must directly address the prior interpretations that Mlu is a differentiated subpopulation within Msa, rather than a distinct species, or provide evidence for why it should be considered a distinct species.

Indeed, some literatures revealed that *Miscanthus lutarioriparius* is a subspecies or ecotype of *Miscanthus sacchariflorus* (Sun et al. 2010; Clark et al. 2019), but we clarified that *M. lutarioriparius* is an endemic species in central China based on its distribution, morphology, physiological trait and cytogenetic karyotypes, etc.

Based on the flora of China (Chen and Renvoize, 2006), *Miscanthus lutarioriparius* and *Miscanthus sacchariflorus* are considered to be two distinct species. The full name of a species contains information about changes in the taxonomic understanding of that species: *Miscanthus lutarioriparius* L. Liu ex Renvoize & S. L. Chen. And we think the view of point that *Miscanthus lutarioriparius* is a native species of China has been widely approved (Chen and Renvoize, 2006; Sheng et al., 2016; Yang et al., 2019).

There are significant differences between *M. lutarioriparius* and *M. sacchariflorus* in morphology, geographical distribution, cytogenetic karyotypes, photosynthesis:

First, *M. lutarioriparius* is always 3-7 m tall and has branches at nodes. While *M. sacchariflorus* could only grow 65-160 cm tall, and has no branch at nodes (Chen and Renvoize, 2006).

Second, *M. lutarioriparius* is mainly distributed in the rivers alongside the middle and lower reaches of the Yangtze River (Xi, 2000; Yan et al., 2016; Li et al., 2016). While *M. sacchariflorus* are mainly distributed in north China (Li et al., 2016).

Recently, a molecular cytogenetic characterization study of four *Miscanthus* species revealed that the karyotype of *M. sacchariflorus* is the most symmetrical, and that of *M. lutarioriparius* is the most asymmetrical (Tang et al., 2019). Tang et al. (2019) used molecular cytogenetic karyotypes to effectively distinguish the two species. They also reported that 45S rDNA heterozygosity in *M. sacchariflorus* but not in *M. lutarioriparius*.

Forth, when transplanted *M. lutarioriparius* and *M. sacchariflorus* in common gardens at different latitudes, researchers found the two species had significant differentiations about photosynthetic rate and water use efficiency. Following transcriptome analyses confirmed the intrinsic mechanisms from the insight of gene controls (Yan et al. 2014; Fan et al. 2015; Xing et al. 2016).

The photos of four *Miscanthus* species taken by our co-author's group (Laigeng Li) show significant differences in morphology with distinct leaf and straw compositions for aboveground biomass (Fig. 4) (Liu et al., 2013). Obviously, *M. lutarioriparius* owns highest straw-to-leaf ratio (89.2/10.8), while *M. sacchariflorus* owns a lower ratio (42.6/57.4) (Liu et al., 2013). Additionally, the chemical composition of *M. lutarioriparius* is also different from *M. sacchariflorus*. And their research indicated that there was significant difference in biomass production among the four species, among which *M. lutarioriparius* had the highest annual dry biomass production (32.0 t/ha) while *M. sacchariflorus* had only 16.7 t/ha (Liu et al., 2013). That is the reason why researchers in China put considerable attention on *M. lutarioriparius* as an energy crop rather than *M. sacchariflorus*. For example, professor Tao Sang, one of our corresponding authors, performed a lot of researches on stressful environments adaption of *M. lutarioriparius* (Mi et al., 2014; Xing et al., 2016; Zhu et al., 2017; Yan et al., 2017; Wang et al., 2019).

Fig. 4 Plant phenotypes of four *Miscanthus* species.

The photos were taken in September after four *Miscanthus* species have fully developed. Bar = 1 m. Figure is from the research paper of our co-author Laigeng Li (Liu et al., 2013).

We would like to emphasize that the biomass of *M. lutarioriparius* is significantly higher than that of other major Chinese *Miscanthus* species, which is an important basis for *M. lutarioriparius* as native energy crop in China.

The methods provide no details about the Mlu plant that was sequenced other than it was collected from Honghu city. The authors should provide more details on the sequenced plant (akin to a voucher that would be deposited in a germplasm collection).

The sample collection process is as follows: Firstly, when sampling in Honghu Lake, Hubei Province, China, the sample was determined according to morphological differences, and the plant was dug back and planted in Wuhan Botanical Garden. After a period of growth, the root tip was taken for karyotype and flow cytometry analysis to determine the genome size and diploid of the plant. After that, the individual was propagated through asexual reproduction. This individual sequenced in this study was selected from Honghu Lake, Hubei Province, because of its vigorous growth and large population size in local.

Another major claim of the manuscript is that the Mlu genome harbors a significantly larger proportion of tandemly duplicated sequences compared to seven other plant genomes, which may be associated with evolution and adaptation of *Miscanthus*. This interpretation is supported by the predicted functions enriched among tandemly

duplicated genes, which includes cell wall synthesis, photosynthesis, stress responses, and pollen response, traits that might enhance fitness in a broadly adapted perennial with a self-incompatible breeding system.

The implication from Figure 3 is that higher gene family members is a response (retention) to phenotypic selection and adaptation. However, one caution to the gene family analysis is that among the 7 plant species compared for Figure 3, Mlu is the only one of these to have experienced a recent likely allopolyploidy event, (Ssp is autopolyploid, Zm diploidized ancient tetraploid, others all diploids). Thus, the gene family size may be inflated in Mlu because the whole genome duplication has not yet progressed far along the path to diploidization compared to the other species. The observation that many of these duplicated genes are not highly expressed (a sign of pseudogenization), raises the possibility that they may not be major contributors to plant function. Perhaps these interpretations could be strengthened by an evolutionary analysis of some of these genes (e.g. the CA genes) among a broader diversity of *Miscanthus*?

Thanks. The recent WGD does have big impact on the expansion of gene family size. It is an import source for gene family size expansion. While the WGD has little direct influence on the occurrence of tandemly duplicated genes, since the mechanisms are different. Function of tandemly duplicated genes showed enrichment in biotic and abiotic stress response in rice and *Arabidopsis* (Rizzon et al., 2006). In our study, we also found the tandemly duplicated genes enriched in biotic and abiotic stress response function, providing confidence in our analysis. Additionally, the tandemly duplicated genes of *M. lutarioriparius* enriched in the cell wall biosynthesis, which is associated with the distinct trait of *Miscanthus*—larger biomass production.

In this manuscript, we carefully analyzed the transcriptome expression pattern of gene families of C4 photosynthesis pathway. We found most WGD duplicates of putative C4 genes have high and leaf-specific expression, while the other non-C4 isoform duplicates had low expression level among all transcriptome samples. It should be emphasized that *Miscanthus* has higher efficient C4 photosynthesis under cool or cold condition compared with other C4 plant, such as maize. Wang et al. revealed that increased PPDK RNA transcription and/or the stability of this RNA are important for the increase in PPDK protein content and activity in *M. x giganteus* under chilling conditions relative to maize (Wang et al., 2007). The high and leaf-specific expression of C4 WGD duplicates in *M. lutarioriparius* probably increase the RNA transcription of related C4 genes, which still need experiment evidence to confirm. However, the influence of WGD on the expansion of gene family size cannot be denied.

1. Lines 37-39 of the Abstract, the meaning here is not clear. Most of the gene duplicates are not expressed, so then probably not functional? Are these CNV/PAVs?

Thanks. In order to avoid ambiguity, we deleted this part of content.

We analyzed the gene families involved in C4 photosynthesis pathway, cellulose biosynthesis, lignin biosynthesis, and found that most duplicated genes in these gene families had relatively low level of expression. For instance, there were 10 CA genes identified in *M. lutarioriparius* genome, most of which originated from tandem duplication. Only two genes MI06G028470, MI05G027550 were highly and specifically expressed in leaf, while the rest CA genes showed relatively low expression level in all transcriptome samples. Considering that leaves are the main site of C4 photosynthesis and collinear relationship with sorghum C4 isoform, we

believed that MI06G028470, MI05G027550 are the functional CA genes. However, without experimental verification, we can't be sure these genes with low expression level are nonfunctional.

PAV and CNV are referred to genomic polymorphisms at the interspecific or intraspecific levels. Here, we have no resequencing samples from *M. lutarioriparius* species or genome assembly of other *M. lutarioriparius* individuals, which is why we did not explore the PAVs and CNVs in *M. lutarioriparius*, so we are not sure whether these duplicated genes are CNVs or PAVs. CNVs in plants have been proved to affect domestication traits (Diaz et al., 2012; Lye and Purugganan, 2019). So, in the near future, we maybe perform further research about CNV/PAV in *M. lutarioriparius*.

Line 46, “the chilling even cold, which made” should be “chilling, even cold, which makes”.

Thanks. We have changed the content as advised.

Reference

- Chen, S, and Renvoize, SA. *Miscanthus* Andersson, Öfvers. Kongl. Vetensk.-Akad. Förh. 12: 165. 1855.
- Li, X, Liao, H, Fan, C, et al. Distinct geographical distribution of the *Miscanthus* accessions with varied biomass enzymatic saccharification. *PLoS One*. 2016, **11**: e0160026.
- Liu, C, Xiao, L, Jiang, J, et al. Biomass properties from different *Miscanthus* species. *Food Energy Secur.* 2013, **2**: 12–19.
- Clark, L V., Jin, X, Petersen, KK, et al. Population structure of *Miscanthus sacchariflorus* reveals two major polyploidization events, tetraploid-mediated unidirectional introgression from diploid *M. sinensis*, and diversity centred around the Yellow Sea. *Ann Bot.* 2019, **124**: 731–748.
- Diaz, A, Zikhali, M, Turner, AS, et al. Copy number variation affecting the photoperiod-b1 and vernalization-a1 genes is associated with altered flowering time in wheat (*Triticum aestivum*). *PLoS One*. 2012, **7**: e33234.
- Fan, Y, Wang, Q, Kang, L, et al. Transcriptome-wide characterization of candidate genes for improving the water use efficiency of energy crops grown on semiarid land. *J Exp Bot.* 2015, **66**: 6415–6429.
- Lye, ZN, and Purugganan, MD. Copy number variation in domestication. *Trends Plant Sci.* 2019, **24**: 352–365.
- Mi, J, Liu, W, Yang, W, et al. Carbon sequestration by *Miscanthus* energy crops plantations in a broad range semi-arid marginal land in China. *Sci Total Environ.* 2014, **496**: 373–380.
- Paterson, A. H. et al. The *Sorghum bicolor* genome and the diversification of grasses. *Nature* 2009, **457**, 551–556.
- Rizzon, C, Ponger, L, and Gaut, BS. Striking similarities in the genomic distribution of tandemly arrayed genes in *Arabidopsis* and rice. *PLoS Comput Biol.* 2006, **2**: e115.
- Sheng, J, Hu, X, Zeng, X, et al. Nuclear DNA content in *Miscanthus* sp. and the geographical variation pattern in *Miscanthus lutarioriparius*. *Sci Rep.* 2016, **6**: 34342.
- Sun, Q, Lin, Q, Yi, ZL, et al. A taxonomic revision of *Miscanthus* s.l. (Poaceae) from China. *Bot J Linn Soc.* 2010, **164**: 178–220.
- Tang, Y-M, Xiao, L-QLL-Q, Iqbal, Y, et al. Molecular cytogenetic characterization and phylogenetic analysis of four *Miscanthus* species (Poaceae). *Comp Cytogenet.* 2019, **13**: 211–230.
- Wang, Q, Kang, L, Lin, C, et al. Transcriptomic evaluation of *Miscanthus* photosynthetic traits to salinity stress. *Biomass and Bioenergy* 2019, **125**: 123–130.

Wang, X., Tang, H. & Paterson, A. H. Seventy million years of concerted evolution of a homoeologous chromosome pair, in parallel, in major Poaceae lineages. *Plant Cell* **23**, 27–37 (2011).

Xi, Q. Investigation on the distribution and potential of giant grasses in China: *Triarrhena*, *Miscanthus*, *Arundo*, *Phragmites* and *Neyraudia*.

Xing, S, Kang, L, Xu, Q, et al. The coordination of gene expression within photosynthesis pathway for acclimation of C₄ energy crop *Miscanthus lutarioriparius*. *Front Plant Sci.* 2016, **7**.

Yan, J, Zhu, M, Liu, W, et al. Genetic variation and bidirectional gene flow in the riparian plant *Miscanthus lutarioriparius*, across its endemic range: implications for adaptive potential. *GCB Bioenergy*. 2016, **8**: 764–776.

Yan, J, Song, Z, Xu, Q, et al. Haplotypes phased from population transcriptomes detecting selection in the initial adaptation of *Miscanthus lutarioriparius* to stressful environments. *Mol Ecol*. 2017, **26**: 5911–5922.

Yang, S, Xue, S, Kang, W, et al. Genetic diversity and population structure of *Miscanthus lutarioriparius*, an endemic plant of China. *PLoS One*. 2019, **14**, e0211471.

Reviewer #2 (Remarks to Author):

In the manuscript entitled “Chromosome-scale assembly and analysis of biomass crop *Miscanthus lutarioriparius* genome”, Miao and colleagues assembled chromosome-scale *Miscanthus lutarioriparius* genome based on Nanopore sequences and HiC, explored the genome structures, analyzed several key gene families associate with biomass and estimated the divergence of *Miscanthus*. As I known, the quality of current genome is much improved assembly than the available *Miscanthus* genome so far in despite of the potential redundancy assembly. However, in comparison with the recent published genome projects, the study only conducted very basic analysis for the genome based on de novo sequences and limited RNA-seq. In addition, *Miscanthus* genome has already be available. Thus, the study only can provide very limited novel discovery for genome evolution and the genetic basis of biomass in *Miscanthus lutarioriparius*. And I don't think the current manuscript meet standard requirements for Nature Communication.

Thanks for the comments on the quality of the *Miscanthus lutarioriparius* genome assembly. We believe that the biomass of *M. lutarioriparius* is significantly higher than that of other major Chinese *Miscanthus* species, and that the genome assembly of *M. lutarioriparius* will provide genetic basis for energy crop improvement. In addition, this study has confirmed the allotetraploid origin of *M. lutarioriparius* using the centromeric satellite repeats. We also used our genome assembly as a reference sequence to investigate the genetic diversity of *M. lutarioriparius* populations in China with transcriptome data. The comparative genomics analysis between *Miscanthus lutarioriparius* and *Miscanthus sinensis* was also performed, through which we assigned 19 chromosome of *M. lutarioriparius* into two sub-genomes. These data have been added into the revised manuscript.

Some general comments/suggestions on the analyses conducted:

1. Line 190-192 “2,802 (8.21%) sorghum genes have more than two syntenic genes in *M. lutarioriparius* genome, which presumably resulted from the segmental duplication, tandem duplication or single gene duplication occurred

in *M. lutarioriparius* genome after it split with sorghum.” This phenomenon is usually caused by the heterozygosity of the genome since *Miscanthus* is self-incompatible. Hereinafter, the authors had the conclusions that” the heterozygosity in *Miscanthus* genome hindered the advances of the genome sequencing and assembly in the past”. The authors may try to use the relative bioinformatic tools (for example: `purge_haplotigs` (https://bitbucket.org/mroachawri/purge_haplotigs/src/master/) to remove the potential heterozygosity assembly and then manually check the reassembled genome.

We understand that using some software to remove the redundant sequence will make the assembly perfect. We tried `purge_haplotigs` with different parameters as recommended by reviewer. The detailed process is described as following:

1. `minimap2 -ax map-ont contig.fasta corrected_reads.fasta --secondary=no | samtools sort -@ 12 -m 4G -o aligned.bam &`
2. `purge_haplotigs hist -b aligned.bam -g contg.fasta &`
3. `purge_haplotigs contigcov -i aligned.bam.gencov -l 0 -m 46 -h 115`
4. `purge_haplotigs purge -g contig.fasta -c coverage_stats.csv &”`

We finally got 1.875 Gb total length, which is significantly smaller than the genome size estimated by flow cytometry. The default percent cutoff for identifying a contig as a haplotig is 70. Then we tried 80 with following command: `purge_haplotigs purge -g contig.fasta -c coverage_stats.csv -a 80`. When we increased the percent cutoff for identifying a contig as a haplotig from 70 to 80, the total length of the reassembled genome changed from 1.875 Gb to 2.044 Gb. We are concerned that highly repetitive sequences and tandem/segmental duplication blocks in our *M. lutarioriparius* genome will lead to much more contigs identified as haplotigs with default parameters. However, it is hard to know which parameter is closer to the truth.

We tried to look into the extent of sequence redundancy containing in our assembly through the syntenic depth analysis of *M. lutarioriparius* and sorghum, which was previously described in our manuscript: About 87.05% (29,710) of sorghum genes have 2 syntenic genes in *M. lutarioriparius*, 2,802 (8.21%) sorghum genes have more than two syntenic genes in *M. lutarioriparius* genome.

So, we checked 2,159 of 2,802 sorghum genes that has more than two *M. lutarioriparius* syntenic genes (Why 2,159? Since these 2,159 genes are located in the seed syteny blocks (High quality)). We put the raw file `MI.Sb.anchors` generated by JCVI (MCScan python version) on figshare: <https://doi.org/10.6084/m9.figshare.13297880>. Detailed, 2,159 sorghum genes are collinear with 6,843 *M. lutarioriparius* genes. Among these 6,843 *M. lutarioriparius* genes, 620 genes are located on the scaffolds that failed in chromosome scaffolding process. These 620 genes probably be generated by heterozygosity. But the most common case (1,367/2,159) is that two are WGD duplicates, and the third is usually physically close to one of the two WGD duplicates. For instance, `Sobic.001G009800` are collinear with `MI01G073970`, `MI01G074210` and `MI17G036850`. And `MI01G07397` is very close to `MI01G074210` in physical position.

Of the 2,159 sorghum genes investigated, 211 sorghum genes contained 4 syntenic *M. lutarioriparius* genes. We carefully checked these 211 sorghum genes and found 68 of them have three/two *M. lutarioriparius* syntenic genes located in chromosomes, and one/two *M. lutarioriparius* syntenic genes located in scaffolds,

which may represent heterozygosity. However, 143 sorghum genes have four syntenic *M. lutarioriparius* genes that located on two chromosomes and the two genes are physically close together, which probably originate from gene duplication before the recent WGD. We put the file that containing information about sorghum genes having 4 *M. lutarioriparius* syntenic genes in figshare: <https://doi.org/10.6084/m9.figshare.13297883.v1>.

Therefore, we believe that the main reason for this phenomenon (2,802 (8.21%) sorghum genes have more than two syntenic genes in *M. lutarioriparius* genome) is gene duplication rather than heterozygosity. So, we didn't take the reassembly generated by `purge_haplotigs`.

At the beginning of the project, we worried about heterozygosity in *M. lutarioriparius* genome, because its self-incompatible. And we noticed that DOE-JGI constructed a *Miscanthus sinensis* DH (double haploid) plant for genome sequencing and assembly (https://phytozome.jgi.doe.gov/pz/portal.html#!info?alias=Org_Msinensis_er), which could eliminate impact of heterozygosity for genome assembly. Fortunately, the genome size of our preliminary contig assembly is a little bigger than that estimated by flow cytometry, indicating the heterozygosity in our sequencing individual is very low. Later we used the Illumina sequencing data to assess the heterozygosity by mapping the reads against the final genome assembly and calculating the percentage of heterozygous SNPs in genome. The result showed a relatively low heterozygosity SNPs percentage, 0.65%.

Recently, the DOE-JGI published their paper (Mitros et al., 2020), so the limitation of usage of *Miscanthus sinensis* DH genome has been unlocked. The predicted gene numbers of *M. sinensis* and *M. lutarioriparius* are very close (67,967 versus 68,328), providing confidence for our result.

2. Line 204-207: “Besides, inter-chromosomal exchanges between MIChr09/10 and MIChr14/15 were observed, which probably occurred prior to the *M. lutarioriparius* recent WGD (indicated with green circles in Fig. 2a).” The tip of two chromosomes shared high collinearly in major Poaceae lineages, which is caused by concerted evolution of a homoeologous chromosome pair. The authors should review the relative publications---Paterson, A. H. et al. Nature 457, 551 (2009). Wang, X., Tang, H. & Paterson, A.. Plant Cell 23, 27–37 (2011). Wang, X et al, have estimated that the gene conversions were occurred ~13Mya in sorghum. The authors may estimate the divergence time by themselves based on the *Ks* of homologous gene pairs.

Thank the reviewer for recommending us to read the relative publications. We realized that we have misinterpreted this phenomenon. Therefore, we have modified our description by “The ends of MIChr09/10 and MIChr14/15 are highly collinear, which common to major Poaceae lineages (Paterson et al., 2009; Wang et al., 2011) (indicated with green circles in Fig. 3a).

We looked into the result of gene synteny analysis for *M. lutarioriparius* and *S. bicolor*. The detailed number of gene pairs in the green circles (Fig. 2a) is described as follows:

- MIChr14 and SbChr05: 66 high quality collinear gene pairs;
- MIChr15 and SbChr05: 66 high quality collinear gene pairs;
- MIChr09 and SbChr08: 42 high quality collinear gene pairs;
- MIChr10 and SbChr08: 73 high quality collinear gene pairs.

MIChr09 and MIChr10 have 38 collinear gene pairs at chromosome tips.

MIChr14 and MIChr15 have 44 collinear gene pairs at chromosome tips.

The above 82 (38+44) pairs of genes are collinear with sorghum in the green circles indicated in manuscript Fig. 2.

We calculated the K_s values of these 82 homologues gene pairs, and the divergence time of these homologues genes was estimated to be ~8.15 MYA (Fig. 5).

Fig. 5 Homology between ends of chromosome 9, 10, 14 and 15

Lines between chromosomes connect syntenic genes, and colors correspond to K_s values

3. C4 pathway genes of *M. lutarioriparius* could be an interesting topic, I did not see any evidence to support the C4 characteristics (probably the gene expression pattern of leaf?) of these genes beside the orthologues analysis. Since the WGD were occurred in *M. lutarioriparius* ~ 6 MYA, two copies derived from the WGD may be divergence for their functions including C4 characteristics. The author may refer to Li, P., (2010. Nature Genetics 42, 1060) and Pick, T.R. et al. (2011, Plant Cell 23, 4208.) for the identification of C4 genes.

Thanks. The characterization of C4 genes in *M. lutarioriparius* will be more reliable if the expression pattern based on the transcriptomes of bundle sheath and mesophyll cells are available. Since the identification of most C4 genes in sorghum and maize has the expression pattern evidence of bundle sheath and mesophyll cells. For instance, sorghum and maize C4 PPCK genes' C4 nature/characteristics are supported by evidence that their expression is light-induced, and their transcripts are more abundant in mesophyll than bundle-sheath cells (Shenton et al., 2006). The sorghum C4 PEPC gene (Sb10g021330) was characterized to have more than 20 times more abundant in mesophyll than in bundle-sheath cells (Wyrich et al., 1998). The transcript of sorghum C4 gene Sb03g003230 was abundant in bundle-sheath, but not in mesophyll cells (Wyrich et al., 1998). Anyway, the expression of C4 genes are located in leaves. In our study, expect for *M. lutarioriparius* C4 pathway genes which is collinear with sorghum C4 gene, the rest genes have very low expression level in leaves. Based on the orthologue's analysis with sorghum C4 genes and leaf-specific high expression pattern, we have great

confidence of identification of C4 genes in *M. lutarioriparius*. The high and leaf-specific expression pattern is an important evidence to support the C4 characteristics.

The two C4 genes originated from WGD have very similar expression pattern with specific high expression in leaf. Therefore, there is insufficient evidence to infer functional differentiation of WGD C4 gene in *M. lutarioriparius*. It should be emphasized that *Miscanthus* has higher efficient C4 photosynthesis under cold or cool condition compare other C4 plant, such as maize. Wang et al. revealed that increased PPDK RNA transcription and/or the stability of this RNA are important for the increase in PPDK protein content and activity in *M. x giganteus* under chilling conditions relative to maize (Wang et al., 2008). Therefore, we speculate the functional WGD C4 genes may contribute to the cool C4 photosynthesis in *Miscanthus*.

Some minor issues:

1. Fig. 1a. Why the chromosome ID in both the X-axis and Y-axis were not based on the order? It is probably accorded to the length of pseudo chromosomes?

After Hi-C scaffolding, we renamed the scaffolds based on the syntenic relationship with sorghum chromosomes. That is the reason why chromosome ID in Hi-C heatmap were not based on the order.

The legend for this figure –“The green circles indicate the inter-chromosomal exchanges among the chromosome 441 ends of MIChr09, MIChr10, MIChr14 and MIChr15, which probably happened before the *M. lutarioriparius* specific WGD.” is not appropriate. Please see my comment above.

Thanks. We have already modified that description by “The green circles indicate that the chromosome ends of MIChr09/10 and MIChr14/15 are highly collinear, which common to major Poaceae lineages.”.

2. “The genome size estimated by k-mer statistics is about 2.19 Gb, close to 2.147 Gb determined using flow cytometry”. The significant figures should be consistent.

Thanks, we have already changed 2.147 into 2.15.

3. Line: 123: 99,78% should be 99.78%

Thanks, we have already modified it.

4. Some of figures in Supplementary are unreadable, for examples: Supplementary Fig. 26, Supplementary Fig. 32a,

Thanks. Sorry for that. To avoid overlapping words, we chose smaller fonts, which might make it impossible to read gene names clearly. Therefore, we decided to provide supplementary tables containing gene information to enable readers to make better use of the results. Please see supplementary table 17 and 22.

Reference:

Mitros, T, Session, AM, James, BT, et al. Genome biology of the paleotetraploid perennial biomass crop *Miscanthus*. *Nat Commun.* 2020, **11**: 5442.

Paterson, A. H. et al. The *Sorghum bicolor* genome and the diversification of grasses. *Nature* 2009, **457**, 551–556.

Shenton, M, Fontaine, V, Hartwell, J, et al. Distinct patterns of control and expression amongst members of the PEP carboxylase kinase gene family in C₄ plants. *Plant J.* 2006, **48**: 45–53.

Wang, D, Portis, AR, Moose, SP, et al. Cool C₄ photosynthesis: pyruvate pi dikinase expression and activity corresponds to the exceptional cold tolerance of carbon assimilation in *Miscanthus × giganteus*. *Plant Physiol.* 2008, **148**: 557–567.

Wang, X., Tang, H. & Paterson, A. H. Seventy million years of concerted evolution of a homoeologous chromosome pair, in parallel, in major Poaceae lineages. *Plant Cell* **23**, 27–37 (2011).

Wyrich, R, Dressen, U, Brockmann, S, et al. The molecular basis of C₄ photosynthesis in sorghum: isolation, characterization and RFLP mapping of mesophyll- and bundle-sheath-specific cDNAs obtained by differential screening. *Plant Mol Biol.* 1998, **37**: 319–335.

Reviewer #3

This is an excellent investigation, executed with the appropriate techniques and presented in a clear and concise manner. All of the methods were appropriate, and the results have been thoroughly and carefully evaluated. This project provides valuable new information that will be useful for plant genomics research, especially in areas concerning the biology of grasses.

The only criticism I have of the manuscript is that it needs to be carefully edited for improved wording. The organization and content are excellent, but the writing has numerous grammatical errors. These should be relatively easy to correct. I have indicated a substantial number of revisions on the scanned copy included in this review but recommend that the manuscript be carefully revised for grammar.

Many thanks for the comments and suggestions. We have revised the manuscript and added some new contents into the revision. In addition to correcting the grammar errors pointed out by reviewers, we also corrected some other errors using traceable mode in our revised manuscript.

Additionally, we added a section named “Centromeric evolution in *Miscanthus lutarioriparius*”, which confirmed the allotetraploid origin of *M. lutarioriparius* using the centromeric satellite repeats. We also used our genome assembly as a reference sequence to investigate the genetic diversity of *M. lutarioriparius* populations with transcriptome data. This part was added into the section named “Genome evolutionary history and genetic diversity analysis of *Miscanthus lutarioriparius*”. Comparative genomics analysis was also performed between *Miscanthus lutarioriparius* and *Miscanthus sinensis*.

REVIEWER COMMENTS

Reviewer #1 (Remarks to the Author):

The authors have submitted an improved manuscript that addresses many of the comments from the prior review. An important issue is that there was no comparison of the *lutarioriparius* (Mlu) genome assembly to the *M. sinensis* (Msi) genome assembly that has been available at Phytozome since 2017. The authors' response is that they did not include these analyses because the Msi genome assembly was released under the "Fort Lauderdale" agreement, where genome-scale analyses are not authorized until the genome is published. This is only partially true, as such analyses were in fact possible if the authors had requested approval from the Msi genome consortium, which included researchers from all over the globe and thus open to collaboration. Ignoring the obvious question of comparisons among the Mlu and Msi assemblies is questionable ethics at best, and certainly not good science.

Fortunately, the issue is moot now as the Msi genome has been published, and the authors now make some basic comparisons of these two *Miscanthus* genomes. As expected, they are highly collinear, but two specific comments on this analysis follow.

1. How was the sequence similarity (47.67% genome-wide) in Supp Table 26 calculated, and how are these values to be interpreted? The nucleotide sequence identity in coding sequences between *Miscanthus sacchariflorus* (Msa) and Msi is greater than 95%, and although divergence in non-coding sequence is expected, it is not expected to be below 50%. We know from other grass genomes that there may be substantial presence-absence variation and changes in repeat element organization that would reduce pairwise sequence identity, but a chromosome-scale value offers little insight into its biological basis.
2. The authors point out there are a number of small inversions between the Mlu and Msi genome assemblies, which they say could be due to assembly error or species differences. Because the Msi genome assembly was verified by high density genetic mapping, but the Mlu genome assembly was not, the authors should qualify this statement by stating these differences may be due to errors in their Mlu assembly, rather than Msi.

The authors were also asked to provide evidence for why Mlu should be considered a distinct species, instead of simply an ecotype of Msa. The response is basically, "because other people (mostly botanists) say so in prior publications", based on morphological descriptions or physiological traits. Genome sequencing has revealed that prior taxonomic classifications into separate groups are not always supported by molecular evidence. In fact, the recent Msi genome paper analyzed genomic diversity within the *Miscanthus* genus and showed that the previously named *M. transmorrisonensis* and *M. floridulus* are likely Msi subtypes, and that due to extensive admixture, even some *M. sacchariflorus* have been misclassified. Furthermore, although Mlu does have distinct phenotypes from other Msa, it is possible that those phenotypes could arise from single gene mutations of pleiotropic effect, such as genes controlling photoperiod sensitivity, flowering time, dwarfing genes, etc. So, what this reviewer wants to know is, now that there is a complete genome assembly for Mlu, what is the GENOMIC SEQUENCE EVIDENCE that supports Mlu as a distinct species, and not just a locally-adapted variant of tetraploid *M. sacchariflorus*? Line 416-417 state that these relationships will be clearer if more genomes are included, and that data is now available through the recent Msi genome publication.

The authors continue to emphasize the functional importance of local tandem gene duplications, despite the likely possibility that some and perhaps many of these instances could represent heterozygosity. This possibility can be directly addressed by investigating how many of these local tandem gene duplications are also in the Msi genome, where the impact of heterozygosity has been eliminated because a doubled haploid plant was sequenced. If also present in Msi, then it would be more appropriate to say that these tandem duplications could contribute to phenotypes in the

Miscanthus genus more broadly, instead of only the distinctive features of Mlu.

In addressing this question in the revision, the authors also make a curious statement that based on limited expansion of their contig assembly relative to estimated genome size, the Mlu individual they sequenced has low heterozygosity. I actually doubt this is true, considering Mlu is self-incompatible and the transcriptome data they use to describe variation within Mlu populations in Supplementary Figure 25 shows substantial genetic diversity (very few individuals on the same branch). I don't remember if this Mlu diversity analysis was included in the original manuscript submission, but regardless, since it is presented now, what is the relationship of the individual plant they sequenced for genome assembly to these populations? Does it belong in Group I or Group II?

The claim of tandem duplicate genes contributing to adaptive phenotypes in Mlu is not supported by any direct evidence, and is rather an inference from GO term enrichments and expanded gene families. The authors are cautioned to consider the possibility that the structural features of certain types of genes, with leucine-rich repeat genes being an obvious example, make them prone to more rapid evolution by tandem duplication, especially when they occur in gene clusters.

Reviewer #2 (Remarks to the Author):

The authors have improved the revised manuscript and my previous comments have been addressed. However, a number of issues remain that I think will need further attention.

1. In Abstract, "The 2.07-Gb assembly covers 96.64% of the genome" don't make any sense since the genome size was estimated.

2. Figures,

Fig 2b is unreadable, please improve.

Fig 3b, the authors may label the peak values for the Ks of the two WGD. The ancient WGD is supposed to be the rho p WGD (Refer to Ming, et al., 2015, Nature Genetics)

3. Discussion

The discussion should be improved. The contents in this section are conclusions rather than discussions. Particularly, Line 419-423, I did not see any discussion for the phylogeny. The authors may compare their results with previous study and try to reach any conclusion/hypothesis.

Line 401: "The recent whole genome duplication" should be "The recent WGD".

Round #2

REVIEWER COMMENTS

Reviewer #1 (Remarks to the Author):

The authors have submitted an improved manuscript that addresses many of the comments from the prior review. An important issue is that there was no comparison of the *M. lutarioriparius* (Mlu) genome assembly to the *M. sinensis* (Msi) genome assembly that has been available at Phytozome since 2017. The authors' response is that they did not include these analyses because the Msi genome assembly was released under the "Fort Lauderdale" agreement, where genome-scale analyses are not authorized until the genome is published. This is only partially true, as such analyses were in fact possible if the authors had requested approval from the Msi genome consortium, which included researchers from all over the globe and thus open to collaboration. Ignoring the obvious question of comparisons among the Mlu and Msi assemblies is questionable ethics at best, and certainly not good science.

Fortunately, the issue is moot now as the Msi genome has been published, and the authors now make some basic comparisons of these two *Miscanthus* genomes. As expected, they are highly collinear, but two specific comments on this analysis follow.

1. How was the sequence similarity (47.67% genome-wide) in Supp Table 26 calculated, and how are these values to be interpreted? The nucleotide sequence identity in coding sequences between *Miscanthus sacchariflorus* (Msa) and Msi is greater than 95%, and although divergence in non-coding sequence is expected, it is not expected to be below 50%. We know from other grass genomes that there may be substantial presence-absence variation and changes in repeat element organization that would reduce pairwise sequence identity, but a chromosome-scale value offers little insight into its biological basis.

2. The authors point out there are a number of small inversions between the Mlu and Msi genome assemblies, which they say could be due to assembly error or species differences. Because the Msi genome assembly was verified by high density genetic mapping, but the Mlu genome assembly was not, the authors should qualify this statement by stating these differences may be due to errors in their Mlu assembly, rather than Msi.

The authors were also asked to provide evidence for why Mlu should be considered a distinct species, instead of simply an ecotype of Msa. The response is basically, "because other people (mostly botanists) say so in prior publications", based on morphological descriptions or physiological traits. Genome sequencing has revealed that prior taxonomic classifications into separate groups are not always supported by molecular evidence. In fact, the recent Msi genome paper analyzed genomic diversity within the *Miscanthus* genus and showed that the previously named *M. transmorrisonensis* and *M. floridulus* are likely Msi subtypes, and that due to extensive admixture, even some *M. sacchariflorus* have been misclassified. Furthermore, although Mlu does have distinct phenotypes from other Msa, it is possible that those phenotypes could arise from single gene mutations of pleiotropic effect, such as genes controlling photoperiod sensitivity,

flowering time, dwarfing genes, etc. So, what this reviewer wants to know is, now that there is a complete genome assembly for Mlu, what is the GENOMIC SEQUENCE EVIDENCE that supports Mlu as a distinct species, and not just a locally adapted variant of tetraploid *M. sacchariflorus*? Line 416-417 state that these relationships will be clearer if more genomes are included, and that data is now available through the recent Msi genome publication.

The authors continue to emphasize the functional importance of local tandem gene duplications, despite the likely possibility that some and perhaps many of these instances could represent heterozygosity. This possibility can be directly addressed by investigating how many of these local tandem gene duplications are also in the Msi genome, where the impact of heterozygosity has been eliminated because a doubled haploid plant was sequenced. If also present in Mis, then it would be more appropriate to say that these tandem duplications could contribute to phenotypes in the *Miscanthus* genus more broadly, instead of only the distinctive features of Mlu.

In addressing this question in the revision, the authors also make a curious statement that based on limited expansion of their contig assembly relative to estimated genome size, the Mlu individual they sequenced has low heterozygosity. I actually doubt this is true, considering Mlu is self-incompatible and the transcriptome data they use to describe variation within Mlu populations in Supplementary Figure 25 shows substantial genetic diversity (very few individuals on the same branch). I don't remember if this Mlu diversity analysis was included in the original manuscript submission, but regardless, since it is presented now, what is the relationship of the individual plant they sequenced for genome assembly to these populations? Does it belong in Group I or Group II?

The claim of tandem duplicate genes contributing to adaptive phenotypes in Mlu is not supported by any direct evidence, and is rather an inference from GO term enrichments and expanded gene families. The authors are cautioned to consider the possibility that the structural features of certain types of genes, with leucine-rich repeat genes being an obvious example, make them prone to more rapid evolution by tandem duplication, especially when they occur in gene clusters.

Reviewer #2 (Remarks to the Author):

The authors have improved the revised manuscript and my previous comments have been addressed. However, a number of issues remain that I think will need further attention.

1. In Abstract, “The 2.07-Gb assembly covers 96.64% of the genome” don't make any sense since the genome size was estimated.

2. Figures,

Fig 2b is unreadable, please improve.

Fig 3b, the authors may label the peak values for the Ks of the two WGD. The ancient WGD is supposed to be the rho ρ WGD (Refer to Ming, et al., 2015, Nature Genetics)

3. Discussion

The discussion should be improved. The contents in this section are conclusions rather than discussions. Particularly, Line 419-423, I did not see any discussion for the phylogeny. The authors may compare their results with previous study and try to reach any conclusion/hypothesis.

Line 401: “The recent whole genome duplication” should be “The recent WGD” .

Round #2 Response

REVIEWER COMMENTS

Reviewer #1 (Remarks to the Author):

The authors have submitted an improved manuscript that addresses many of the comments from the prior review. An important issue is that there was no comparison of the *M. lutarioriparius* (Mlu) genome assembly to the *M. sinensis* (Msi) genome assembly that has been available at Phytosome since 2017. The authors’ response is that they did not include these analyses because the Msi genome assembly was released under the “Fort Lauderdale” agreement, where genome-scale analyses are not authorized until the genome is published. This is only partially true, as such analyses were in fact possible if the authors had requested approval from the Msi genome consortium, which included researchers from all over the globe and thus open to collaboration. Ignoring the obvious question of comparisons among the Mlu and Msi assemblies is questionable ethics at best, and certainly not good science.

Fortunately, the issue is moot now as the Msi genome has been published, and the authors now make some basic comparisons of these two *Miscanthus* genomes. As expected, they are highly collinear, but two specific comments on this analysis follow.

Answer:

Thanks.

Here, we would like to briefly introduce our *Miscanthus lutarioriparius* genome sequencing project. The project was started more than ten years ago. In 2009, Illumina paired-end library and mate-pair library were chosen as the sequencing strategy. Because the continuity and quality of that *de novo* assembly did not meet our expectations, PacBio long reads were later introduced to scaffold, which improved our previous assembly but still short from generating good enough results. Since PacBio sequencing was expensive and the sequencing data quality was not good enough at that time, we got very limited reads. Thanks to the great advance in the third-generation sequencing, reads were getting longer with decreased costs. More recently, we were able to use Oxford Nanopore sequencing in combination with Hi-C technology and achieved very good results.

The final completion of the long-term effort to sequence the *M. lutarioriparius* genome was by itself satisfactory to us because it was the effort that has witnessed the evolution of sequencing technologies. When we noticed that *M. sinensis* genome data was released under the “Fort Lauderdale” agreement, we did not rush to finish our project but considered to conduct a comparative genomic analysis when *M. sinensis* genome was published at any point in future. As suggested by reviewer, we performed the comparative genomic analysis between *M. lutarioriparius* and *M. sinensis*. We assigned 19 *M. lutarioriparius* chromosomes into two sub-genomes based on the wonderful work did by Msi genome consortium. Obviously, there are much more studies waiting to be done by the bioenergy community when these genome sequences became available.

As a team having participated in the International Rice Genome Sequencing Project, we have benefited tremendously from the past and present international collaborations. We have finished several genomes sequencing projects, such as the accurate sequence of rice chromosome 4 (Feng et al., 2002), *de novo* whole genome assembly of bamboo (Peng et al., 2013) and grass carp (Wang et al., 2015). We did not involve the collaboration with *M. sinensis* genome consortium this time, since we already started sequencing *M. lutarioriparius*, an endemic species in Central China. Despite the lack of continuing funding for the bioenergy projects, we tried our best to bring in funding from various sources and insisted the completion of the project. This has been the project that lasted for the longest time and experienced the most numerous phases of technical advances. Thanks to the *M. sinensis* genome consortium, the opportunity for integration of these *Miscanthus* genome assemblies will provide more genomic resource for the future intensively comparative genomic studies of energy plants.

Reference:

Feng, Q., Zhang, Y.J., Hao, P., *et al.* Sequence and analysis of rice chromosome 4. *Nature*. 420:316–320 (2002).

Peng, Z.H., Lu, Y., Li, L.B., *et al.* The draft genome of the fast-growing non-timber forest species moso bamboo (*Phyllostachys heterocycla*). *Nat. Genet.* 45:456–461 (2013).

Wang, Y.P., Lu, Y., Zhang, Y., *et al.* The draft genome of the grass carp (*Ctenopharyngodon idellus*) provides insights into its evolution and vegetarian adaptation. *Nat. Genet.* 47: 625–631 (2015).

1. How was the sequence similarity (47.67% genome-wide) in Supp Table 26 calculated, and how are these values to be interpreted? The nucleotide sequence identity in coding sequences between *Miscanthus sacchariflorus* (Msa) and Msi is greater than 95%, and although divergence in non-coding sequence is expected, it is not expected to be below 50%. We know from other grass genomes that there may be substantial presence-absence variation and changes in repeat element organization that would reduce pairwise sequence identity, but a chromosome-scale value offers little insight into its biological basis.

Answer:

Thanks for your valuable advice. Firstly, the whole genome sequence similarity was calculated based on the result of Minimap software. We performed sequence alignment for Mlu and Msi using Minimap software, and the sequence similarity was roughly calculated using the formula: $1 - (\text{accumulated mismatched length} / \text{accumulated alignment length})$.

Here, we recalculated the sequence similarity using the output file generated by MCScan (python version) ([https://github.com/tanghaibao/jcvi/wiki/MCscan-\(Python-version\)](https://github.com/tanghaibao/jcvi/wiki/MCscan-(Python-version))). MCScan calls LAST (version 980) to do the pairwise synteny search, and filtered the LAST output to remove tandem duplications and weak hits. We used the filtered output of LAST (file name: Ml.Msi.last.filter. We then put this file in figshare website of this project) to calculate the sequence similarity of coding regions of *M. lutarioriparius* and *M. sinensis* syntenic gene pairs, which would offer much more insight into the biological basis than the chromosome-scale value we provided previously.

Totally, 33,536 Mlu and Msi syntenic gene pairs, with tandem duplications and weak hits removed, were involved in our analysis. The total alignment length of coding sequences of Mlu and Msi was about 38.99 Mb. The total length of mismatched base was 431,161 bp for coding sequence comparison. The length-weighted average sequence similarity of coding sequence was about 98.59% for Mlu and Msi.

2. The authors point out there are a number of small inversions between the Mlu and Msi genome assemblies, which they say could be due to assembly error or species differences. Because the Msi genome assembly was verified by high density genetic mapping, but the Mlu genome assembly was not, the authors should qualify this statement by stating these differences may be due to errors in their Mlu assembly, rather than Msi.

Answer:

Thanks.

In our round #1 rebuttal letter, we didn't rule out the possibility that these differences might be due to errors in our Mlu assembly.

Here, we'd like to show our observations. For better comparison, we put the synteny dot-plots of *M. lutarioriparius* versus *S. bicolor* and *M. sinensis* versus *S. bicolor* together (Figure 1). We used the same commands and parameters within MCScan (python version) ([https://github.com/tanghaibao/jcvi/wiki/MCscan-\(Python-version\)](https://github.com/tanghaibao/jcvi/wiki/MCscan-(Python-version))) to draw the following two dot graphs. So, these two synteny dot-plots should be comparable. The diagonal of synteny dot-plot of *M. lutarioriparius* versus *S. bicolor* (Figure 1a) is much smoother than that of *M. sinensis* versus *S. bicolor* (Figure 1b). Another synteny analysis comparison also presents similar phenomenon. The synteny dot-plot of *M. lutarioriparius* versus *S. spontaneum* has fewer points than that of *M. sinensis* versus *S. spontaneum* except along the diagonal. And the diagonal of *M. lutarioriparius* versus *S. spontaneum* looks smoother than that of *M.*

sinensis versus *S. spontaneum* (Figure 2). Although the good collinearity with closely relative species (sorghum here) cannot directly reflect the assembly quality and accuracy, it still provide insight into the genome structure variation information.

Figure 1. The gene synteny analysis for *M. lutarioriparius* versus *S. bicolor* and *M. sinensis* versus *S. bicolor*.

Figure 2. The gene synteny analysis for *M. lutarioriparius* versus *S. spontaneum* and *M. sinensis* versus *S. spontaneum*.

The authors were also asked to provide evidence for why Mlu should be considered a distinct species, instead of simply an ecotype of Msa. The response is basically, “because other people (mostly botanists) say so in prior publications, based on morphological descriptions or physiological traits. Genome sequencing has revealed that prior taxonomic classifications into

separate groups are not always supported by molecular evidence. In fact, the recent Msi genome paper analyzed genomic diversity within the *Miscanthus* genus and showed that the previously named *M. transmorrisonensis* and *M. floridulus* are likely Msi subtypes, and that due to extensive admixture, even some *M. sacchariflorus* have been misclassified. Furthermore, although Mlu does have distinct phenotypes from other Msa, it is possible that those phenotypes could arise from single gene mutations of pleiotropic effect, such as genes controlling photoperiod sensitivity, flowering time, dwarfing genes, etc. So, what this reviewer wants to know is, now that there is a complete genome assembly for Mlu, what is the GENOMIC SEQUENCE EVIDENCE that supports Mlu as a distinct species, and not just a locally-adapted variant of tetraploid *M. sacchariflorus*? Line 416-417 state that these relationships will be clearer if more genomes are included, and that data is now available through the recent Msi genome publication.

Answer:

The species concept is indeed one of most elusive problems in biology. Whether a species should be defined based primarily on morphology, ecology, genetics, or genomics has remained controversial for decades. In the case of *M. lutarioriparius*, it is morphologically distinct from *M. sacchariflorus* and stands out as a taxon with the largest biomass production among *Miscanthus* species (Liu et al., 2013; Yang et al., 2019; Xi 2000). It also occupied a unique ecological niche, i.e., seasonally flooded river banks where other plant can hardly adapt. It has also been shown as a valuable genetic resource for bioenergy crop development based on its outstanding photosynthetic rates and water use efficiency when planted in the semiarid marginal land (Yan et al., 2015; Yan et al., 2016; Song et al., 2017; Wang et al., 2019). Therefore, despite the existing controversies, the recognition of its species status would be valuable for its potential utilization as an important entity of genetic resource for bioenergy development.

In the round #1 rebuttal letter, in addition to the morphological and physiological evidence, we did cite certain evidence of molecular cytogenetic characterization. Citation from our round #1 rebuttal letter: “The karyotype of *M. sacchariflorus* is the most symmetrical, and that of *M. lutarioriparius* is the most asymmetrical (Tang et al., 2019). Tang et al. (2019) used molecular cytogenetic karyotypes to effectively distinguish these two species. They also reported that 45S rDNA heterozygosity in *M. sacchariflorus* but not in *M. lutarioriparius*”.

Chae et al., 2014 also provided chromosomal evidence that supported *M. lutarioriparius* was a distinct species rather than an ecotype of *M. sacchariflorus*.

We agree with the reviewer that there might be only a small number of gene differences between *M. lutarioriparius* and *M. sacchariflorus* that lead to the differentiation in their morphology, physiology, and ecology. As indicated by the reviewer, these genes could be involved in photoperiod sensitivity, which would cause different flowering time and consequentially reproductive isolation. Such genes are known as speciation genes. In the case of *M. lutarioriparius*, the key mutation might have led to its adaptation to the unique ecological niches and then geological isolation from the mother populations of *M. sacchariflorus*, which is known as ecological or geological speciation. Thus, the number of

gene difference between taxa may not as a good indication of whether they should be recognized as distinct species. Nevertheless, we agree with the reviewer that the availability of these genome sequences will benefit not only our identification of key genes leading to the species differentiation but also the valuable gene resources for energy crop development through either QTL mapping or GWAS. We however leave this kind of extensive and in-depth studies for the future researchers or colleagues specialized in these areas because they would be much more authoritative than us in carrying out such studies.

Reference:

- Chae, W.B., Hong, S.J., Gifford, J.M., *et al.* Plant morphology, genome size, and SSR markers differentiate five distinct taxonomic groups among accessions in the genus *Miscanthus*. *GCB Bioenergy*. 6:646–660 (2014).
- Feng, Q., Zhang, Y.J., Hao, P., *et al.* Sequence and analysis of rice chromosome 4. *Nature*. 420:316–320 (2002).
- Liu, C., Xiao, L., Jiang, J., *et al.* Biomass properties from different *Miscanthus* species. *Food Energy Secur.* 2:12–19 (2013).
- Song, Z. *et al.* Transcriptomic characterization of candidate genes responsive to salt tolerance of *Miscanthus* energy crops. *GCB Bioenergy*. 9:1222–1237 (2017).
- Tang, Y.M., Xiao, L., Iqbal, Y., *et al.* Molecular cytogenetic characterization and phylogenetic analysis of four *Miscanthus* species (Poaceae). *Comp Cytogenet.* 13: 211–230 (2019).
- Wang, Q., Kang, L., Lin, C., Song, Z., Tao, C., Liu, W., Sang, T., and Yan, J. Transcriptomic evaluation of *Miscanthus* photosynthetic traits to salinity stress. *Biomass and Bioenergy*. 125:123–130 (2019).
- Yan, J., Zhu, C., Liu, W., Luo, F., Mi, J., Ren, Y., Li, J., and Sang, T. High photosynthetic rate and water use efficiency of *Miscanthus lutarioriparius* characterize an energy crop in the semiarid temperate region. *GCB Bioenergy*. 7:207–218 (2015).
- Yan, J., Zhu, M., Liu, W., Xu, Q., Zhu, C., Li, J., and Sang, T. Genetic variation and bidirectional gene flow in the riparian plant *Miscanthus lutarioriparius*, across its endemic range: implications for adaptive potential. *GCB Bioenergy*. 8:764–776 (2016).
- Xi Q. Investigation on the distribution and potential of giant grasses in China: *Triarrhena*, *Miscanthus*, *Arundo*, *Phragmites* and *Neyraudia*. 2000.
- Yang, S., Xue, S., Kang, W., *et al.* Genetic diversity and population structure of *Miscanthus lutarioriparius*, an endemic plant of China. *PLoS One*. 14:1–18 (2019).

The authors continue to emphasize the functional importance of local tandem gene duplications, despite the likely possibility that some and perhaps many of these instances could represent heterozygosity. This possibility can be directly addressed by investigating how many of these local tandem gene duplications are also in the Msi genome, where the impact of heterozygosity has been eliminated because a doubled haploid plant was sequenced. If also present in Msi, then it would be more appropriate to say that these tandem duplications could contribute to phenotypes in the *Miscanthus* genus more broadly, instead of only the distinctive features of Mlu.

Answer:

Thanks for your questions and valuable advice.

Here, we used the output file (MI.Msi.lifted.anchors) generated by MCScan (python version) to investigate the syntenic genes of 4,365 *M. lutarioriparius* tandem duplicated genes in *M. sinensis* genome. (Note: the file MI.Msi.lifted.anchors recruits additional anchors to form the final synteny blocks, please see the file description on [https://github.com/tanghaibao/jcvi/wiki/MCscan-\(Python-version\)#dependencies](https://github.com/tanghaibao/jcvi/wiki/MCscan-(Python-version)#dependencies)). We put key output files of MCScan in our figshare database for others to reproduce our conclusion.

We found that 4,365 *M. lutarioriparius* tandem duplicated genes have 4,302 syntenic genes in *M. sinensis* using the output file generated by MCScan (python version), indicating that most of the *M. lutarioriparius* tandemly duplicated genes had corresponding syntenic genes in *M. sinensis*. Additionally, the classification of duplicate gene origins of *M. sinensis* showed that 3,797 (5.60%) genes of *M. sinensis* were classified as tandem duplicates, which was a little less than that of *M. lutarioriparius* (4,365, 6.24%).

Thus, it would be more appropriate to speculate that these tandem duplications could have contribute to phenotypes such as stress response and biomass production in the *M. lutarioriparius*.

In addressing this question in the revision, the authors also make a curious statement that based on limited expansion of their contig assembly relative to estimated genome size, the Mlu individual they sequenced has low heterozygosity. I actually doubt this is true, considering Mlu is self-incompatible and the transcriptome data they use to describe variation within Mlu populations in Supplementary Figure 25 shows substantial genetic diversity (very few individuals on the same branch). I don't remember if this Mlu diversity analysis was included in the original manuscript submission, but regardless, since it is presented now, what is the relationship of the individual plant they sequenced for genome assembly to these populations? Does it belong in Group I or Group II?

Answer:

Here, we did rough heterozygosity assessment by mapping the Illumina short reads (~41X sequencing depth) back to the assembly. A total of 11,044,595 heterozygous SNPs and 483,392 InDels were identified in our *M. lutarioriparius* genome. An overall heterozygous rate of the occurrence of SNPs and InDels was estimated at about 5.6 polymorphism per kilobase, which was a little more than that (4.2 per kilobase) of grape genome (Velasco et al., 2007), more than that (2.6 per kilobase) of poplar genome (Tuskan et al., 2006) and that (1.0 per kilobase) of moso bamboo (Peng et al., 2013).

Yes, this part of Mlu diversity analysis was not included in the original manuscript submission. Here, we analyzed our sequences with the previous published transcriptome data of *M. lutarioriparius* to determine where our *M. lutarioriparius* stands among natural populations. About 3.3 M SNPs were identified from 80 individual's transcriptome data and sequence data obtained by our project. When we filtered these variants with some basic conditions (the missing rate of individual genotypes is requested to be less than 20%, and the minimum allele frequency (MAF) should be greater than 1%), 781,089 SNPs were kept for the phylogeny analysis. The phylogenetic analysis indicated that the *M. lutarioriparius* individual sequenced in this study belonged to group II in the resulting phylogeny (Figure 3).

Figure 3. Neighbor-joining tree reconstructed using the SNP identified by transcriptome data. Branches of different colors indicate different populations of *Miscanthus lutarioriparius*. The dotted lines in green and blue represent Group I and Group II, respectively.

Reference:

- Tuskan, G. A., DiFazio, S., Jansson, S., Bohlmann, J., Grigoriev, I., Hellsten, U., Putnam, M., Ralph, S., Rombauts, S., Salamov, A., *et al.* (2006). The genome of black cottonwood, *Populus trichocarpa* (Torr. & Gray). *Science*. 313(5793):1596–1604 (2006).
- Velasco, R., Zharkikh, A., Troggio, M., Cartwright, D. A., Cestaro, A., Pruss, D., Pindo, M., FitzGerald, L. M., Vezzulli, S., Reid, J., *et al.* A high quality draft consensus sequence of the genome of a heterozygous grapevine variety. *PLoS One* 2:e1326 (2007).
- Peng, Z.H., Lu, Y., Li, L.B., *et al.* The draft genome of the fast-growing non-timber forest species moso bamboo (*Phyllostachys heterocycla*). *Nat. Genet.* 45:456–461 (2013).

The claim of tandem duplicate genes contributing to adaptive phenotypes in Mlu is not supported

by any direct evidence and is rather an inference from GO term enrichments and expanded gene families. The authors are cautioned to consider the possibility that the structural features of certain types of genes, with leucine-rich repeat genes being an obvious example, make them prone to more rapid evolution by tandem duplication, especially when they occur in gene clusters.

Answer:

Thanks for your question and kind reminder.

The tandem duplicate genes contributing to the adaptative phenotypes were claimed in rice and *Arabidopsis* in the previous study (Rizzon et al., 2006), which gave us reference to draw the conclusion on *M. lutarioriparius* tandem duplicates. Furthermore, in addition to the duplicated genes related to adaptative phenotypes, those related to cell wall biosynthesis were also enriched for tandem duplicates in *M. lutarioriparius*.

Yes, leucine-rich repeat genes, especially NBS-LRR encoding genes did seem to play a role in adaption.

Reference:

Rizzon, C., Ponger, L., and Gaut, B.S. Striking similarities in the genomic distribution of tandemly arrayed genes in *Arabidopsis* and rice. *PLoS Comput Biol.* 2: e115 (2006).

Reviewer #2 (Remarks to the Author):

The authors have improved the revised manuscript and my previous comments have been addressed. However, a number of issues remain that I think will need further attention.

1. In Abstract, “The 2.07-Gb assembly covers 96.64% of the genome” don't make any sense since the genome size was estimated.

Answer:

Thanks.

The genome size of our sequencing individual was estimated using flow cytometry by our co-authors in their previous publication (Li, 2013). We cited the data of *M. lutarioriparius* genome size and calculated the coverage of 96.64% (Please see our citation in Line 114-115: The final assembly covers 96.64% of 2,147 Mb genome on the basis of the flow cytometry¹²). Using the genome size estimated by flow cytometry as the gold standard is acceptable. However, we did not make it clear as it should be in the previous version of the manuscript, which has been corrected in the revised manuscript.

Reference:

12. Li, X. *et al.* Nuclear DNA content variation of three *Miscanthus* species in China. *Genes and Genomics.* 35:13–20 (2013).

2. Figures,
Fig 2b is unreadable, please improve.

Answer:

Thanks for your reminder. We have modified the Fig 2b by adding the tree scale number 1.

Fig 3b, the authors may label the peak values for the Ks of the two WGD. The ancient WGD is supposed to be the rho ρ WGD (Refer to Ming, et al., 2015, Nature Genetics)

Answer:

Thanks. We have labeled the peak values in Fig. 3b. The numbers in parentheses indicate the peak values of recent *M. lutarioriparius* WGD and the grass lineage shared ρ WGD.

3. Discussion

The discussion should be improved. The contents in this section are conclusions rather than discussions. Particularly, Line 419-423, I did not see any discussion for the phylogeny. The authors may compare their results with previous study and try to reach any conclusion/hypothesis.

Answer:

Thanks. We added our discussion about the phylogeny of *Miscanthus* as following: “In the case of *M. lutarioriparius*, it is morphologically distinct from *M. sacchariflorus* and stands out as a taxon with the largest biomass production among *Miscanthus* species^{11,58}. It also occupied a unique ecological niche, i.e., seasonally flooded riverbanks where other plant can hardly adapt. It has also been shown as a valuable genetic resource for bioenergy crop development based on its outstanding performance of photosynthetic rates and water use efficiency when planted in the semiarid marginal land^{13,59–61}. Phylogeny based on chloroplast genome sequence provides a new perspective for the interspecific relationship of genus *Miscanthus*. Therefore, despite the existing controversies, the recognition of its species status would be valuable for its potential utilization as an important unit of genetic resource for bioenergy development.”

Reference:

Chae, W.B., Hong, S.J., Gifford, J.M. *et al.* Plant morphology, genome size, and SSR markers differentiate five distinct taxonomic groups among accessions in the genus *Miscanthus*. *GCB Bioenergy*. 6: 646–660 (2014).

Liu, C., Xiao, L., Jiang, J., *et al.* Biomass properties from different *Miscanthus* species. *Food Energy Secur.* 2:12–19 (2013).

Song, Z. *et al.* Transcriptomic characterization of candidate genes responsive to salt tolerance of *Miscanthus* energy crops. *GCB Bioenergy*. 9:1222–1237 (2017).

Wang, Q., Kang, L., Lin, C., Song, Z., Tao, C., Liu, W., Sang, T., and Yan, J. Transcriptomic evaluation of *Miscanthus* photosynthetic traits to salinity stress. *Biomass and Bioenergy*. 125:123–130 (2019).

Yan, J., Zhu, C., Liu, W., Luo, F., Mi, J., Ren, Y., Li, J., and Sang, T. High photosynthetic rate and water use efficiency of *Miscanthus lutarioriparius* characterize an energy crop in the semiarid temperate region. *GCB Bioenergy*. 7:207–218 (2015).

Yan, J., Zhu, M., Liu, W., Xu, Q., Zhu, C., Li, J., and Sang, T. Genetic variation and bidirectional gene flow in the riparian plant *Miscanthus lutarioriparius*, across its endemic range: implications for adaptive potential. *GCB Bioenergy*. 8:764–776 (2016).

Xi Q. Investigation on the distribution and potential of giant grasses in China: *Triarrhena*, *Miscanthus*, *Arundo*, *Phragmites* and *Neyraudia*. 2000.

Yang, S., Xue, S., Kang, W., *et al.* Genetic diversity and population structure of *Miscanthus lutarioriparius*, an endemic plant of China. *PLoS One*. 14:1–18 (2019).

Line 401: “The recent whole genome duplication” should be “The recent WGD” .

Answer:

Thanks. We have modified that.

REVIEWERS' COMMENTS

Reviewer #1 (Remarks to the Author):

The authors have carefully considered the comments offered on the prior revision, and have further strengthened the paper.

This reviewer certainly understands the challenges with sequencing the complex genome of *Miscanthus*, and how it took progress in sequencing technologies and bioinformatics to finally succeed. I believe the effort to sequence *Miscanthus sinensis* began in 2008, so I suspect that group could tell a similar story. Fortunately, due to the persistence of both teams, now the research community will have multiple high quality genome assemblies for *Miscanthus*.

I also agree with the authors about the potential controversies around defining a species, the added discussion on this point is well-reasoned. It seems the genome sequence comparisons alone do not indicate obvious structural variants that would define the possible species within the *Miscanthus* genus, so either the distinguishing genomic features are more subtle, or ecological factors and adaptation may be the primary drivers of the distinct *Miscanthus* types.

The authors now also present a more compelling case for a relatively high degree of tandem duplications in *Miscanthus* genomes, as most of those identified in *M. lutarioriparius* were also observed in *M. sinensis*. However, this observation also suggests that these tandem duplications may contribute primarily to the unique phenotypes across the *Miscanthus* genus, rather than the differences between *Miscanthus* species.

REVIEWERS' COMMENTS

Reviewer #1 (Remarks to the Author):

The authors have carefully considered the comments offered on the prior revision and have further strengthened the paper.

This reviewer certainly understands the challenges with sequencing the complex genome of *Miscanthus*, and how it took progress in sequencing technologies and bioinformatics to finally succeed. I believe the effort to sequence *Miscanthus sinensis* began in 2008, so I suspect that group could tell a similar story. Fortunately, due to the persistence of both teams, now the research community will have multiple high quality genome assemblies for *Miscanthus*.

I also agree with the authors about the potential controversies around defining a species, the added discussion on this point is well-reasoned. It seems the genome sequence comparisons alone do not indicate obvious structural variants that would define the possible species within the *Miscanthus* genus, so either the distinguishing genomic features are more subtle, or ecological factors and adaption may be the primary drivers of the distinct *Miscanthus* types.

The authors now also present a more compelling case for a relatively high degree of tandem duplications in *Miscanthus* genomes, as most of those identified in *M. lutarioriparius* were also observed in *M. sinensis*. However, this observation also suggests that these tandem duplications may contribute primarily to the unique phenotypes across the *Miscanthus* genus, rather than the differences between *Miscanthus* species.

RESEPONSE TO REVIEWERS' COMMENTS

Thanks for your comments and affirmation of our work. Your questions and suggestions have helped us improve a lot of the content of our article. And we also would like to appreciate all other reviewers for their questions and suggestions.